# STRUCTURAL ADVERSARIAL OBJECTIVES FOR SELF-SUPERVISED REPRESENTATION LEARNING

## ABSTRACT

Within the framework of generative adversarial networks (GANs), we propose objectives that task the discriminator with additional structural modeling responsibilities. In combination with an efficient smoothness regularizer imposed on the network, these objectives guide the discriminator to learn to extract informative representations, while maintaining a generator capable of sampling from the domain. Specifically, we influence the features produced by the discriminator at two levels of granularity. At coarse scale, we impose a Gaussian assumption encouraging smoothness and diversified representation, while at finer scale, we group features forming local clusters. Experiments demonstrate that augmenting GANs with these self-supervised objectives suffices to produce discriminators which, evaluated in terms of representation learning, compete with networks trained by state-of-the-art contrastive approaches. Furthermore, operating within the GAN framework frees our system from the reliance on data augmentation schemes that is prevalent across purely contrastive representation learning methods.

## 1 INTRODUCTION

Unsupervised feature learning algorithms aim to directly learn representations from data without reliance on annotations, and have become crucial to efforts to scale vision and language models toward handling real-world complexity. Many state-of-the-art approaches adopt a contrastive self-supervised framework, wherein a deep neural network is tasked with mapping augmented views of a single example to nearby positions in a high-dimension embedding space, while separating embeddings of different examples (Wu et al., 2018; He et al., 2020; Chen et al., 2020; Chen & He, 2021; Grill et al., 2020; Zbontar et al., 2021). Though requiring no annotation, and hence unaffected by assumptions baked into any labeling procedure, the invariances learned by these models are still influenced by human-designed heuristic procedures for creating augmented views.

The recent prominence of contrastive approaches was both preceded by and continues alongside a focus on engineering domain-relevant proxy tasks for self-supervised learning. For computer vision, examples include learning geometric layout (Doersch et al., 2015), colorization (Zhang et al., 2016; Larsson et al., 2017), and inpainting (Pathak et al., 2016; He et al., 2022). Basing task design on domain knowledge may prove effective in increasing learning efficiency, but strays further from an alternative goal of developing truly general and widely applicable unsupervised learning techniques.

A third family of approaches, coupling data generation with representation learning, may provide a path toward such generality while also escaping dependence upon the hand-crafted elements guiding data augmentation or proxy task design. Generative adversarial networks (GANs) (Goodfellow et al., 2020) and variational autoencoders (VAEs) (Kingma & Welling, 2013) are prime examples within this family. Considering GANs, one might expect the discriminator to act as an unsupervised representation learner, driven by the need to model the real data distribution in order to score the generator's output. Indeed, prior work finds that some degree of representation learning occurs within discriminators in a standard GAN framework (Radford et al., 2015). Yet, to improve generator output quality, limiting the capacity of the discriminator appears advantageous (Arjovsky et al., 2017) – a choice potentially in conflict with representation learning. Augmenting the standard GAN framework to separate encoding and discrimination responsibility into different components (Donahue et al., 2017; Dumoulin et al., 2017), along with scaling to larger models (Donahue & Simonyan, 2019), is a promising path to circumventing this apparent limitation.

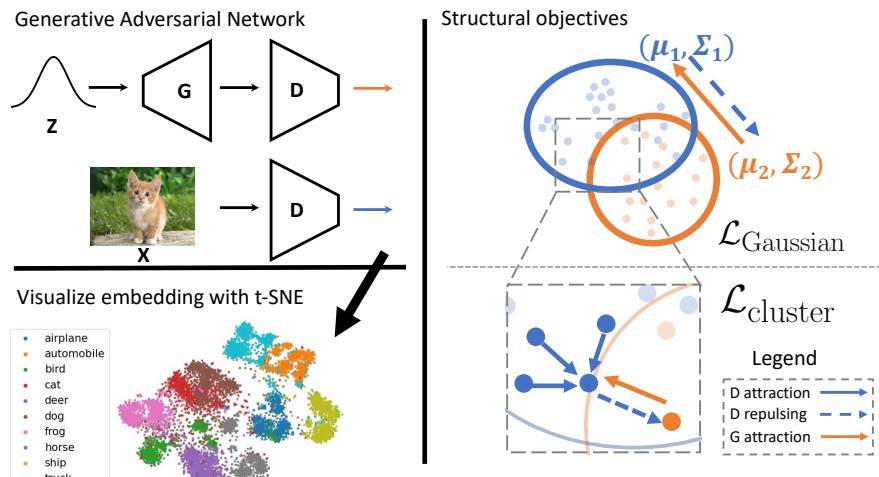

Figure 1: Within the GAN framework, we illustrate our proposed structural adversarial objectives, which push the discriminator to learn good feature representations. We combine objectives from two levels of granularity. *Top right*: At a coarse scale, we model features under the Gaussian assumption, and adversarially orient generated embedding (orange) towards the real embedding (blue), using the sample mean $\mu$ and covariance $\Sigma$. *Bottom right*: At a finer scale, we implement clustering objectives, which group adjacent real embeddings to form a cluster and adversarially attract the generated embedding towards the nearby cluster center. *Bottom left*: Learned representations on CIFAR-10 show a distribution consistent with categorization.

However, it has been unclear whether the struggle to utilize vanilla GANs as effective representation learners stems from inherent limitations of the framework. We provide evidence to the contrary, through an approach that significantly improves representations learned by the discriminator, while maintaining generation quality and operating with a standard pairing of generator and discriminator components. Our approach only modifies the training objectives within the GAN framework, with two aims: (1) regularize the smoothness of the discriminator while maintaining adequate capacity, and (2) require the discriminator to model additional structure of the real data distribution.

To control discriminator smoothness, we propose an efficient regularization scheme that approximates the spectral norm of the Jacobian. Specifically, it instantiates the matrix-vector subroutine in a power iteration with an efficient vector-Jacobian product protocol, preventing the need to compute the full Jacobian matrix of the neural network.

To prompt the discriminator to learn embeddings that model the data distribution, we propose adversarial objectives resembling a contrastive self-supervised clustering target. These objectives influence the output of the discriminator at two levels of granularity. At a coarse level, we make a Gaussian assumption aiming for diversified and smooth representation. At a more refined scale, we adopt a mean shift clustering objective to group features. Though imposing attractive and repulsive forces onto embeddings produced by the discriminator (Figure 1), we do not rely on data augmentation to drive learning. Instead, both real and fake data participate in an adversarial clustering game.

Experiments on representation learning benchmarks show our method achieves competitive performance with recent state-of-the-art contrastive self-supervised learning approaches, even though we do not leverage information from (or even have a concept of) an augmented view. We also demonstrate that supplementing a GAN with our proposed objectives not only enhances the discriminator as a representation learner, but also improves the quality of samples produced by the generator.

## 2 RELATED WORK

### 2.1 GENERATIVE FEATURE LEARNING

Much research on GANs has focused on improving the quality of generated data, yielding significant recent advances (Karras et al., 2017; 2019; 2020; 2021; Sauer et al., 2022). Other efforts have focused on evolving capabilities, including conditional and controllable generation, *e.g.,* text to im-

age generation (Zhang et al., 2021; Hinz et al., 2020) and segmentation-guided image generation (Zhu et al., 2017; Chen & Koltun, 2017). In comparison, adopting GANs for unsupervised feature learning has been relative scarcely explored. In this area, an adversarial approach dependent upon an additional encoder component (Donahue et al., 2017; Dumoulin et al., 2017; Donahue & Simonyan, 2019) appears most successful to date. Here, the encoder is tasked with learning to invert the function of the generator, mapping data to a latent embedding space. The discriminator then acts on (data, latent) pairs, and is tasked with distinguishing between real data paired with the output of the encoder run on it, or vice-versa for a sampled latent and corresponding generator output. Representation learning is the responsibility of the encoder, rather than the discriminator.

Besides GANs, other generative models also demonstrate feature learning capacity: Zhang et al. (2022); Ma et al. (2021) propose to discard the low-level structures in VAE and Flow models to improve learned representations. Du et al. (2021) show that an unsupervised energy model can learn semantic structures, *e.g.,* segmentation and viewpoint, from images. Preechakul et al. (2022) attach an encoder to a diffusion model and show that it learns high-level feature representations.

We adopt an orthogonal approach that, by imposing structural adversarial objectives in a GAN's training, tasks the discriminator to learn richer data representations. Dai et al. (2022) introduce similar objectives in their closed-loop control pipeline, where they replace the pixel-wise reconstruction target of an autoencoder framework with adversarial MCR objectives.

## 2.2 Contrastive Self-Supervised Representation Learning

In contrast to the generative approaches, contrastive self-supervised learning has witnessed substantial progress across multiple domains, sometimes achieving nearly the same performance level as a supervised counterpart. Among them, a popular choice is to learn augmentation invariant representation: Wu et al. (2018); He et al. (2020); Chen et al. (2020); Oord et al. (2018) leverage a siamese pipeline to optimize an InfoNCE objective, which aims to maximize the feature similarity across augmented views, while repulsing from all other instance to maintain feature uniformity. Chen & He (2021); Grill et al. (2020) further simplifies the pipeline by dropping the negative terms and leveraging specific architectural designs to prevent collapsed solutions. As an alternative to operating on an l2 normalized embedding, Caron et al. (2020; 2021); Wang et al. (2021) enforce clustering consistency across views. Inspired by masked language modeling, He et al. (2022); Bao et al. (2021) propose a variant in the image domain by tasking an autoencoder to predict the masked pixels.

Though contrastive approaches yield strong models in multiple benchmarks, Tian et al. (2020) showcase the limitations of view-invariant assumptions and demonstrates their sensitivity to the parameters of augmentation schemes. This implies that the choice of augmentation represents assumptions placed on the data distribution, complicating generalization across multiple data sources. This phenomenon can be observed in the scaling experiment of He et al. (2020), where increasing training images from 1 million to 1 billion leads to only marginal feature improvement. Zhang & Maire (2020) raise a concern with applying those methods to broader and unconstrained datasets, where multiple object instances from the same image should not have mutually invariant representations.

## 2.3 Stabilizing GAN Training

Despite the ability to generate high-quality samples, successfully training GANs remains challenging due to the adversarial optimization. To stabilize the training and scale to larger models, several approaches have been proposed. Heusel et al. (2017) suggest maintaining a separate learning rate for the generator and discriminator to maintain local Nash equilibrium. Arjovsky et al. (2017); Gulrajani et al. (2017) consider constraining the discriminator's Lipschitz constant with gradient clipping and gradient norm penalization. In contrast to regularizing model-wise functionality, Miyato et al. (2018) implement layer-wise spectral normalization schemes by dividing parameters with their leading singular value, which is widely adopted in recent state-of-the-art models. Wu et al. (2021); Bhaskara et al. (2022) instead propose to build a Lipschitz-constrained function by dividing the output with gradient norm and show it can preserve the model capacity. However, none of these methods suit our case, since spectral normalization (Miyato et al., 2018) harms model capacity, and gradient-based regularization only works for scalar output, limiting use of structural objectives.

## 3 METHOD

### 3.1 GAN PRELIMINARIES

Generative adversarial networks (GANs) include two learnable modules: a generator $G$ producing synthetic data from a latent sample $z$, and a discriminator $D$, which learns to differentiate between the true data $x$ and generated samples $G(z)$. During training, $G$ and $D$ are alternatively updated in an adversarial fashion, which can be formulated as a minimax problem (Goodfellow et al., 2020):

$$\min_G \max_D \mathbb{E}_{x \sim p(x)}[\log D(x)] - \mathbb{E}_{\hat{x} \sim g(z)}[1 - \log D(\hat{x})]. \tag{1}$$

There are several popular alternatives for Eqn.1, *e.g.,* hinge loss(Lim & Ye, 2017), Wasserstein distance(Arjovsky et al., 2017), and maximum mean discrepancy (MMD) (Li et al., 2017), but none provide explicit structural control of the output embeddings, *i.e.,* the features produced by $D$. We propose an explicit objective to obtain structural embeddings such that, besides generative power, a well-trained discriminator $D$ accomplishes representation learning.

### 3.2 ADVERSARIAL LEARNING WITH STRUCTURAL OBJECTIVES

Our goal is to task $D$ as the feature extractor to learn data representation on real images. We denote the $p$-dimensional output from $D$ of real images and fake generations as $f$, $f^g$ respectively. Here $f, f^g \in \mathbf{S}^{p-1}$ are normalized and live in a unit hypersphere. We also maintain the unnormalized counterparts $\tilde{f}$ and $\tilde{f}^g$ of $f$ and $f^g$, where we explore their utility in Section 3.3.

Our proposed structural objectives operate hierarchically and regularize the learned embeddings at two levels of granularity: (1) At a coarse level, we assume both $f$, $f^g$ follow Gaussian distributions: $f \sim \mathcal{N}(\mu_f, \Sigma_f), f^g \sim \mathcal{N}(\mu_{f^g}, \Sigma_{f^g})$, where $\mu_f, \mu_{f^g} \in \mathbb{R}^p$ and $\Sigma_f, \Sigma_{f^g} \in \mathbb{R}^{p \times p}$ are sample mean and covariance respectively. (2) At a refined level, we focus on reorganizing embeddings by constructing clusters using local affinity. Therefore, by distributing our objective function with dedicated geometric patterns, we further aim to reinforce the feature learning capability of $D$ under the purely unsupervised setting.

**Coarse-scale Optimization with Gaussian.** Under unimodal Gaussian assumptions, we can simultaneously control the geometry and locations of feature distributions using simple empirical statistics, *i.e.,* sample mean and covariance. With a selected metric $d(\cdot)$ for Gaussian distributions, we define our goal as a minimax problem:

$$\mathcal{L}_{\text{Gaussian}} := \min_G \max_D d(f, f^g). \tag{2}$$

One widely adopted candidate for $d(\cdot)$ is Jensen–Shannon divergence (JSD) due to its symmetry and stability. For two arbitrary probability distributions $P, Q$, JSD admits the following form:

$$\text{JSD}(P\|Q) = D_{\text{KL}}(P\|\frac{P+Q}{2}) + D_{\text{KL}}(Q\|\frac{P+Q}{2}) = H(\frac{P+Q}{2}) - \frac{1}{2}(H(P) + H(Q)), \tag{3}$$

where $D_{\text{KL}}, H$ denotes *Kullback–Leibler divergence* and entropy respectively. We can compute entropy for $Q, P$ using closed-form formulas. However, entropy for $(P + Q)/2$ is problematic to compute exactly and generally requires Monte Carlo simulation, an infeasible computational approach in high dimensional space. To tackle this problem, we follow (Hershey & Olsen, 2007) to approximate $\frac{P+Q}{2}$ by a single Gaussian and estimate sample mean and covariance by unifying the samples of $P$ and $Q$, which yields an upper bound of $H(\frac{P+Q}{2})$ and the bound is tight when $P = Q$. Putting those together, we reach our distance function for coarser scale objectives [1]:

$$\text{JSD}(f, f^g) \approx \log \frac{\det \Sigma_{f+f^g}}{\sqrt{\det \Sigma_f \det \Sigma_{f^g}}}. \tag{4}$$

Another well-established metric between two Gaussian distributions is Bhattacharyya distance $D_B$:

$$D_B(f, f^g) := \frac{1}{8}(\mu_f - \mu_{f^g})^T \Sigma^{-1}(\mu_f - \mu_{f^g}) + \frac{1}{2}\log \frac{\det \Sigma}{\sqrt{\det \Sigma_f \det \Sigma_{f^g}}}, \tag{5}$$

---

[1]Note that though JSD and MCR in Dai et al. (2022) are constructed similarly, the latter computes $X^\top X$ instead of the covariance matrix and is interpreted from coding rate reduction perspective.

where $\mathbf{\Sigma} = \frac{\mathbf{\Sigma}_f + \mathbf{\Sigma}_{f^g}}{2}$. Though having different geometric interpretations, it is notable that $D_B$ and JSD have similar terms, including determinants of the covariance matrix. In the experiments, we observed that these two distances yield similar performance in terms of learned representation, but JSD has a slightly faster convergence rate and better quality for generated images. Therefore, without further specification, we use JSD as our default choice for $d(\cdot)$.

**Fine-grained Optimization with Clustering.** We perform Mean Shift clustering on $\boldsymbol{f}$ by grouping nearby samples. During grouping, we simplify the clustering process by equally averaging each neighbor sample rather than using feature similarity to reweight their contribution. To improve nearest neighbor search stability, we maintain a rolling updated memory bank $\boldsymbol{f}^m$ as a querying pool and use the backbone representation $\boldsymbol{f}^b$ rather than $\boldsymbol{f}$ as the key to computing feature similarity. Denoting $\{\boldsymbol{f}_{i,j}\}_{j=1}^k$ as the returned $K$ nearest neighbors for $\boldsymbol{f}_i$, our clustering objective is:

$$\mathcal{L}_{\text{group}} := \max_D \frac{1}{NK} \sum_{i=1}^N \sum_{j=1}^K \boldsymbol{f}_{i,j}^\top \boldsymbol{f}_i.$$

Note that we do not have a repulsive term here since we empirically find Eqn.2 suffices to maintain feature diversity to prevent collapsed solution. Additionally, we can leverage the clustering objective to perform adversarial optimization by alternatively differentiating and attracting nearby fake features $f^g$ to embed $f^g$ into the clusters eventually. Similarly, with $\{\boldsymbol{f}_{i,j}^g\}_{j=1}^k$ denoted as returned $K$ nearest neighbor real embedding for $f_i^g$, we define an adversarial grouping objective as:

$$\mathcal{L}_{\text{adv group}} := \min_D \max_G \frac{1}{N} \sum_{i=1}^N \sum_{j=1}^K \boldsymbol{f}_{i,j}^{g\top} \boldsymbol{f}_i^g.$$

IC-GAN(Casanova et al., 2021) implements similar instance-wise objective. However, they use an off-the-shelf model rather than jointly optimized embedding to search for the nearest neighbor. During training, we jointly optimize $\mathcal{L}_{\text{cluster}} := \mathcal{L}_{\text{group}} + \mathcal{L}_{\text{adv group}}$ for clustering objectives.

### 3.3 SMOOTHING REGULARIZATION

Besides re-formulating adversarial targets for representation learning, we need to target another common issue in GAN's training: balancing the model's capacity and the smoothness constraint.

Recent studies demonstrate that regularizing $D$'s smoothness, or its Lipschitz constant, is critical for scaling GAN to a giant architecture. Consider a continuous function $F : \mathbb{R}^m \to \mathbb{R}^p$. It is well known that we can bound its Lipschitz constant by the spectral norm of Jacobian $\boldsymbol{J}_F$:

$$\|\boldsymbol{J}_F(\boldsymbol{x})\|_2 \leq \text{Lip},$$

where $\|\cdot\|_2$ denotes the matrix spectral norm. We can efficiently approximate it by adopting the power-iteration (Miyato et al., 2018). However, computing the full Jacobian matrix

---

**Algorithm 1:** Approximating $\|\boldsymbol{J}_F(\boldsymbol{x})\|_2$ with power iterations

---

**Input:** Function $F : \mathbb{R}^m \to \mathbb{R}^p$; Stop gradient operator $sg(\cdot)$; Power iteration steps s; Batch size b; Input data $\boldsymbol{x} \in \mathbb{R}^{b \times m}$;
Initialize random vector $\boldsymbol{u} \sim \mathcal{N}(0,1) \in \mathbb{R}^{b \times p}$
**for** $iter \sim 1 \ldots s$ **do**
 $\boldsymbol{v} = \boldsymbol{u}\boldsymbol{J}_F(\boldsymbol{x})/\|\boldsymbol{u}\boldsymbol{J}_F(\boldsymbol{x})\|_2$  //VJP
 $\boldsymbol{u} = \boldsymbol{J}_F(\boldsymbol{x})\boldsymbol{v}/\|\boldsymbol{J}_F(\boldsymbol{x})\boldsymbol{v}\|_2$  //JVP
**end**
**Return:** $\|\boldsymbol{J}_F(\boldsymbol{x})\|_2 \approx sg(\boldsymbol{u})\boldsymbol{J}_F(\boldsymbol{x})sg(\boldsymbol{v})$

---

is highly inefficient in the current deep learning package where they implement backpropagation using Vector-Jacobian-Product (VJP) subroutine. Each backpropagation only computes a single row of the Jacobian matrix. In our case, it requires $p$ times backpropagation call to compute full $\boldsymbol{J}_D$, which is impracticable as we usually need substantial p to embed representation.

Leveraging the fact that power-iteration is a matrix-free method, we do not need to explicitly compute the Jacobian matrix. Instead, we only need to access the matrix by evaluating the matrix-vector product, which can be efficiently computed by batch-wise VJP and JVP (Jacobian-Vector-Product) subroutine. Algorithm1 presents the details, where only $(2S + 1)$ times of back-propagation are required to approximate $\|\boldsymbol{J}_D(\boldsymbol{x})\|_2$. In our experiments, we find that $S = 1$ suffices for a ResNet-18 model.

We observe that regularizing $\boldsymbol{J}_D(\boldsymbol{x})$ on l2 normalized embedding $\boldsymbol{f}$ will enlarge the norm of $\tilde{\boldsymbol{f}}$ throughout training and eventually destabilizes the system; while regularizing $\boldsymbol{J}_D(\boldsymbol{x})$ on $\tilde{\boldsymbol{f}}$ operates

oppositely. In our method, we choose to compute $\boldsymbol{J}_D(\boldsymbol{x})$ on $\tilde{\boldsymbol{f}}$ and add extra hinge regularization to sustain the embedding norm at a regular level. Therefore, our smoothness regularization is:

$$\mathcal{L}_{\text{reg}} := \min_D \mathbb{E}_{\boldsymbol{x}} \|\boldsymbol{J}_D(\boldsymbol{x}) - \text{Lip}\|_2 + \lambda_{\text{h}} \mathbb{E}_{\tilde{\text{f}}} 1_{\|\tilde{\text{f}}\| \leq 1} \|\tilde{\text{f}} - 1\|_2, \tag{6}$$

where $\lambda_h$ denotes the ratio for hinge regularization, and Lip denotes the Lipschitz target of $D$, which is set to 1 by default. Unlike layer-wise normalization scheme, *e.g.,* Spectral Norm(Miyato et al., 2018), where demanding local regularization hurts the model's capacity; our proposed regularization scheme allows the network to simultaneously fit for multiple objectiveness, *i.e.,* representation learning and smoothness regularization. Therefore, the model does not have to sacrifice capacity for smoothness. Another benefit of our methods is that the proposed term can work with the normalization layer, while Spectral Norm can't because of the data-dependent scaling term in the normalization layer.

**Overall Objective.** Combining all the aforementioned terms, we define our final objective as:

$$\mathcal{L} := \mathcal{L}_{\text{Gaussian}} + \lambda_c \mathcal{L}_{\text{cluster}} + \lambda_s \mathcal{L}_{\text{reg}}, \tag{7}$$

where $\lambda_c, \lambda_s$ control the relative loss weights.

## 4 EXPERIMENTAL SETTINGS

**Datasets.** We conduct experiment on three benchmark datasets: CIFAR-10, CIFAR-100 (Krizhevsky et al., 2009) and ImageNet-10. *ImageNet-10*: we follow (Chang et al., 2017) to select 10 categories from ImageNet dataset (Deng et al., 2009) resulting in 13,000 training images and 500 validation images. During training, we only perform spatial augmentation, including random spatial cropping and horizontal flipping, followed by resizing images to 128x128 resolution to match the generated images. During testing, we resized the images to align the smaller edge to 144, followed by central cropping to produce a 128x128 output. *CIFAR-10/100*: During training, we apply the same augmentation strategy as in ImageNet-10 but produce 32x32 images; During testing, we do not perform cropping.

For compared methods, we keep their default augmentation strategy. In ImageNet-10, we resized ther augmented images to 128x128. We run an unsupervised learning algorithm for all methods on the training split and evaluate the validation split.

**Model Details.**

- *Discriminator*: We construct our discriminator using ResNet-18 (He et al., 2016) and perform several modifications to make it cooperate reasonably with the generator: Inspired by the discriminator configuration in BigGAN, we perform spatial reduction only within the residual block and replace all stride two convolution layers with avg pooling followed by stride one convolution. We remove the first max pooling layer and switch the first convolution layer to a 3x3 kernel with a 1x1 stride to keep the resolution unchanged before the residual block.

  To maintain a substantial down sample rate in ImageNet-10 images, we duplicate the first residual block and enable a spatial reduction in all blocks to reach a 32 down-sample rate. In CIFAR-10/100, we preserve the default setting for residual blocks. As our proposed smoothness term regularizes on each sample, we replace all BatchNorm layer (Ioffe & Szegedy, 2015) with GroupNorm(Wu & He, 2018), specifying 16 channels as a single group, to prevent batchwise interaction. We also remove the first normalization layer in each block as it produces better results. For activation function, we replace ReLU with ELU (Clevert et al., 2015) for broader non-linear support on negative value.

- *Generator*: We adopt the default generator configurations from BigGAN-deep(Brock et al., 2019). Specifically, we take their model for 32x32 images for the experiment in the CIFAR dataset. We additionally increase the base channel to 128 to prevent image generation from being the system bottleneck. For ImageNet-10, we directly take their settings for 128x128 images.

**Optimization Details.** We train our model for 1000 epochs in CIFAR-10/100 and 500 epochs in ImageNet-10. We optimize our objectives using AdamW optimizer (Loshchilov & Hutter, 2017) and use a constant learning rate 2e-4 for both generator and discriminator, We additionally add 0.1 weight decay to the discriminator. We use batch size 500 in CIFAR-10/100 and 320 in ImageNet-10. We run a small-scale parameter tunning experiment for hyperparameters and find that setting

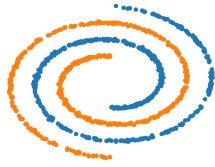 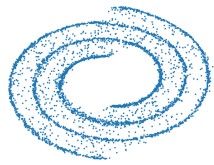 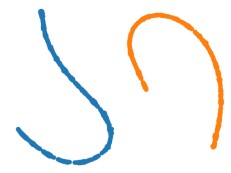 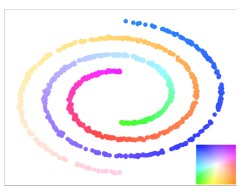

(a) Data samples and categorical assignment

(b) Samples from Generator

(c) Visualize representation with t-SNE

(d) Visualize representation with 2d palette map

Figure 2: We train GAN with our objectives on a synthetic *double spiral* data to qualitatively demonstrate the generated samples and learned representation. (a) Data sample colored by two ground truth categories (b) Generated samples that capture all modes and accurately avoid invalid regions, the space between two double spirals. (c) We use t-SNE to project the learned representation into two-dimensional space and color each point using ground truth assignment. We show that the learned representation correctly separates two categories and preserves continuity within each group. (4) We further showcase the point-wise features by projecting the representation into a 2d palette map, displayed at the bottom right corner, using t-SNE, illustrating that the embedding precisely preserves the point-wise spatial order across and within each group. Best viewed in color.

$\lambda_h, \lambda_c, \lambda_s$ to 4, 3, 5 yields the best result. We run single discriminator updating before optimizing generator, *i.e.*, $n_{dis} = 1$, for simplicity.

In CIFAR10/100 experiment, we use default configurations from SOLO-Learns (da Costa et al., 2022), an open source library providing heavily tuned configurations for multiple state-of-the-art self-supervised methods. In ImageNet-10 experiments, we train the compared approaches using the suggested hyperparameters for ImageNet-100 but extend the total epochs to 1000 for sufficient convergences. We run their methods with our modified backbone and resize input images to 128x128 for a fair comparison.

As a widely adopted trick in GAN's training, we maintain a momentum updated discriminator and generator for evaluation purposes and find it produces stable data representation and better image quality. We also try producing $f^b$ from momentum models for nearest neighbor searching, which brings slightly better performance in all benchmarks.

## 5 RESULTS AND DISCUSSION

### 5.1 SYNTHETIC DATA

Firstly, we demonstrate our learned representation by running our objectives on a synthetic double spirals dataset (Li et al., 2022). In this experiment, we implement discriminator and generator as multi-layer perceptions and keep all other configurations, *e.g.,* normalization layer, choices of activation, objectives function, learning rate, consistent with our settings for real images experiments. In Fig. 2a, we demonstrate that the generated samples precisely capture all data modes, and substantially avoid the outlier samples between spirals. Besides the generation capability, we also visually inspect the learned representations using t-SNE (Van der Maaten & Hinton, 2008) and color each point using ground truth assignment. As shown in Fig.2c, the embedding from the two categories is substantially separated.

In Fig.2d, we further illustrate the detailed structures of the learned representation where we color each data point by projecting learned representation into a 2d palette map, (displayed at the bottom right). From the plot, we can verify the learned embedding preserves relative data structures within and across groups.

### 5.2 REPRESENTATION LEARNING ON REAL IMAGES

We task the backbone of the discriminator to produce a vector as a data representation and then evaluate its performance on the image classification task. We compare the results with state-of-the-art contrastive learning approaches under two widely adopted evaluation metrics:

| Method | CIFAR10 | | CIFAR100 | | ImageNet10 | |
|---|---|---|---|---|---|---|
| | SVM | K-means | SVM | K-means | SVM | K-means |
| Supervised baseline | 95.13 | 95.15 (0.00) | 75.96 | 73.60 (0.78) | 96.40 | 96.37 (0.15) |
| Random | 42.96 | 22.08 (0.09) | 18.39 | 8.99 (0.16) | 48.20 | 28.34 (0.66) |
| NNCLR (Same View) | 29.53 | 20.21(0.22) | 7.70 | 7.25(0.07) | - | - |
| DINO | 89.73 | 63.90 (0.64) | 65.64 | 36.73 (0.47) | 87.80 | 67.98(2.25) |
| NNCLR | 91.69 | 69.32 (3.20) | 69.77 | 40.42 (0.62) | **91.40** | 66.83 (3.20) |
| SimCLR | 90.64 | 75.25 (3.32) | 65.61 | 41.36 (0.59) | 89.00 | 65.73 (2.87) |
| BYOL | **93.08** | 75.03 (2.74) | **70.69** | **42.80** (0.71) | 90.40 | 67.30 (3.34) |
| SWAV | 89.14 | 64.50 (0.91) | 65.07 | 35.26 (0.48) | 90.01 | 61.94 (3.11) |
| Ours | 89.76 | **80.11** (2.84) | 63.34 | 38.16 (0.45) | 91.20 | **75.41** (2.14) |

Table 1: We evaluate our methods by benchmarking the learned representation using linear SVM and K-means clustering. We report averaged accuracy and standard deviation (shown in parentheses), over 20 runs. Our method achieves competitive performance with state-of-the-art self-supervised methods across multiple datasets. However, we do not leverage any augmented views. With NNCLR (Same View), we additionally show removing augmented view information from the NNCLR pipeline yielding degenerated model.

- *Linear Support Vector Machine(SVM)*: We optimize a Linear SVM on top of the l2 normalized training feature and report the classification accuracy on the validation set

- *K-means clustering* : We run spherical K-means clustering on the validation set with K equaling the number of ground truth categories. We then obtain a prediction on the validation set by solving the optimal transport problem between clustering partition and ground truth categories. To reduce the randomness in clustering, we perform five consecutive runs and pick the best clustering result with minimal inertial, *i.e.,*the distance between samples and centroids. We then repeat this process 20 times and report the averaged result and standard deviation. Compared to Linear SVM, K-means clustering evaluates the representation in a more rigorous setting.

We report the results in Table.1 and compare our approach with current state-of-the-art methods. For datasets with fewer categories, *i.e.,*CIFAR-10 and ImageNet-10, our method significantly outperforms the state-of-the-art contrastive learning approaches in K-means clustering. In CIFAR-10, we get 80.11 test accuracy defeating the best-competed method SimCLR with 75.25 test accuracy; In ImageNet-10, our methods reaching 75.41 test accuracy surpasses the best-competed method, DINO with 67.98 test accuracy. When evaluating representation using linear SVM, our method reaches 89.76 test accuracy, which exceeds SWAV, and DINO with 89.14 and 89.73 test accuracy, respectively, but slightly worsens than BYOL with 93.08 and NNCLR with 91.69 test accuracy. In ImageNet10, our method gets 91.20 which approaches the best method NNCLR with 91.40 and exceeds the rest.

In CIFAR-100, there are fewer training samples per category and operationalizing instance-wise discriminating objectives is thus favorable over clustering objectives or smoothness regularization. Under such case, our methods remain competitive on linear SVM metric where we get 63.34 test accuracy, which is very close to DINO, SimCLR, and SWAV, which have around 65 test accuracy. On K-means clustering, our method reaches 38.16 test accuracy outperforming clustering-based contrastive approaches SWAV and DINO, which have 36.73 and 35.26, respectively. In both metrics, our methods are behind the BYOL and NNCLR. We want to point out that, as the value of unsupervised training is on scaling with more data, evaluating self-supervised methods under constrained datasets, as in CIFAR-100, is less informative than in CIFAR-10 and ImageNet-10.

The quantitative comparison can be further qualitatively confirmed by visualizing embedding using t-SNE. BYOL, as shown in Fig. 3a, produces isolated and smaller-sized clusters that maintain sufficient space to discern categories under linear transformation. However, those clusters lack sufficient global-wise organizations, which is evaluated with K-means clustering. On the contrary, our approach, as shown in Fig. 3b and 3c, produces smoother embedding, which are nearly aligned with the ground truth partition yielding good performance on K-means clustering. However, our method's cluster has more outliers, leading to slightly worse linear separability, as measured in

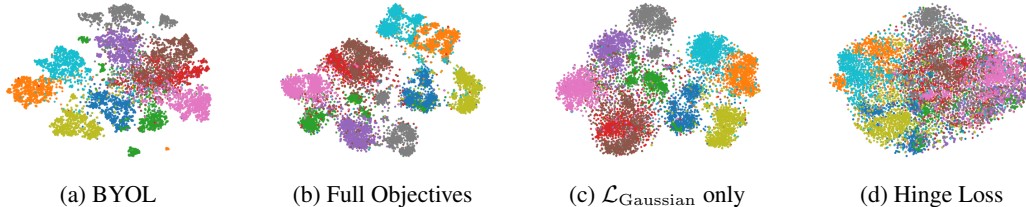

(a) BYOL        (b) Full Objectives        (c) $\mathcal{L}_{\text{Gaussian}}$ only        (d) Hinge Loss

Figure 3: On the CIFAR-10 validation set, we visualize the learned representation using t-SNE and compare with multiple baseline objectives. Training GAN with $\mathcal{L}_{\text{Gaussian}}$ only produces diversified yet over smoothed representation. With cluster loss, the representations from full objectives experiment enhance categorical-wise discrepancy which are visually competitive with state-of-the-art BYOL.

| Method | Inception Scores ↑ | FID ↓ | KMeans | SVM |
|---|---|---|---|---|
| BigGAN(Brock et al., 2019) | 8.22 | 17.50 | 29.69 | 69.31 |
| Hinge Loss | 8.13 | 18.54 | 36.41 | 77.19 |
| $\mathcal{L}_{\text{Gaussian}}$ only, with $D_B$ | 8.39 | 17.83 | 70.76 | 87.9 |
| $\mathcal{L}_{\text{Gaussian}}$ only, with JSD | 8.55 | 16.97 | **80.55** | 88.32 |
| Full objectives | **8.73** | **13.63** | 80.11 | **89.76** |

Table 2: We run ablated experiments and compare different objectives function. Our proposed structural objectives significantly outperform widely adopted hinge loss baseline by a large margin. Including finer scale clustering objectives (last row) improve both representation and image quality over the methods only using coarse scale objectives (3rd and 4th row).

linear SVM. Adding cluster loss will increase the categorical-wise discrimination. Moreover, yield better SVM results.

We run additional baseline experiment by removing the view augmentation from NNCLR pipeline, termed as NNCLR(same view) in Table.1. In this variant, we only align the feature with the nearest neighbor from the memory bank and repulse all other features. Interestingly, such modification breaks the model in all settings and leading even worse results than the randomly initialized baseline. We hypothesize that, without an augmented view providing extra information, nearest neighbor searching does not return useful features, and the model is mostly likely over-fitting noisy artifacts during training.

## 5.3 Image Quality

As shown in Table 2, we provide a quantitative comparison across multiple methods. We show that the proposed objectives significantly improve generation and representation quality over the hinge loss baseline. We can witness further enhancement in image quality with clustering loss that provides extra instance/clustering-wise objectives. Note that the unconditional BigGAN baseline surpasses our hinge loss baseline since BigGAN has a smaller batch size resulting in more updating rounds in the momentum Generator. We provide samples of generated images in the Appendix.

## 6 Conclusion

We propose structural adversarial objectives that augment the GAN framework for self-supervised representation learning. The objectives shape the discriminator's output at two granularity levels: at a coarse scale, features are optimized with Gaussian assumption yielding diversified and smooth representation, while at a finer scale, the nearby features are grouped, forming local clusters. We benchmark across multiple datasets and show supplementing GAN with these self-supervised objectives suffices to produce data representation that is competitive with the state-of-the-art self-supervised learning approaches and substantially improves the quality of generated images.

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

## A  APPENDIX

**Compared Methods** To evaluate the learned representation, we compare our methods with state-of-the-art self-supervised learning methods: SimCLR(Chen et al., 2020) optimizes the InfoNCE loss, maximizing feature similarity across views while repulsing all the images. NNCLR(Dwibedi et al., 2021) sample nearest neighbour from data set using cross view features and treat them as positive for InfoNCE objectives. We additionally run a baseline using NNCLR, removing the augmented view and directly maximizing the similarity between image features and its nearest neighbor. DINO(Caron et al., 2021) and SWAV(Caron et al., 2020) maximizing view consistent objectives using clustering-based targets. BYOL(Grill et al., 2020) only contains the maximizing term and adopts a momentum updated Siamese model to process augmented input to prevent collapsed solution.

For image generation, we compare with the BigGAN(Brock et al., 2019), showing the superior capability to generate complex images. For a fair comparison, we run BigGAN in unconditional mode.

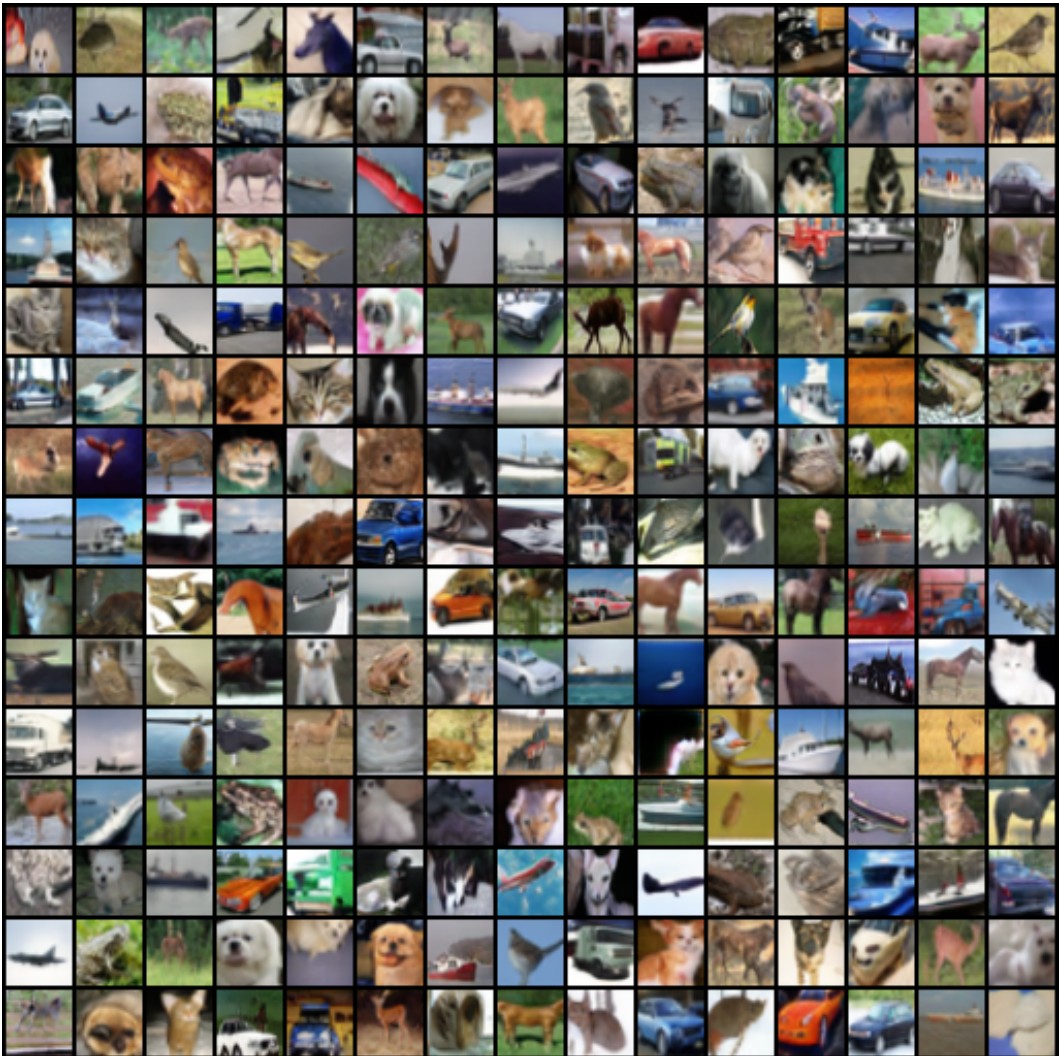

Figure 4: Randomly generated images from our full objectives.

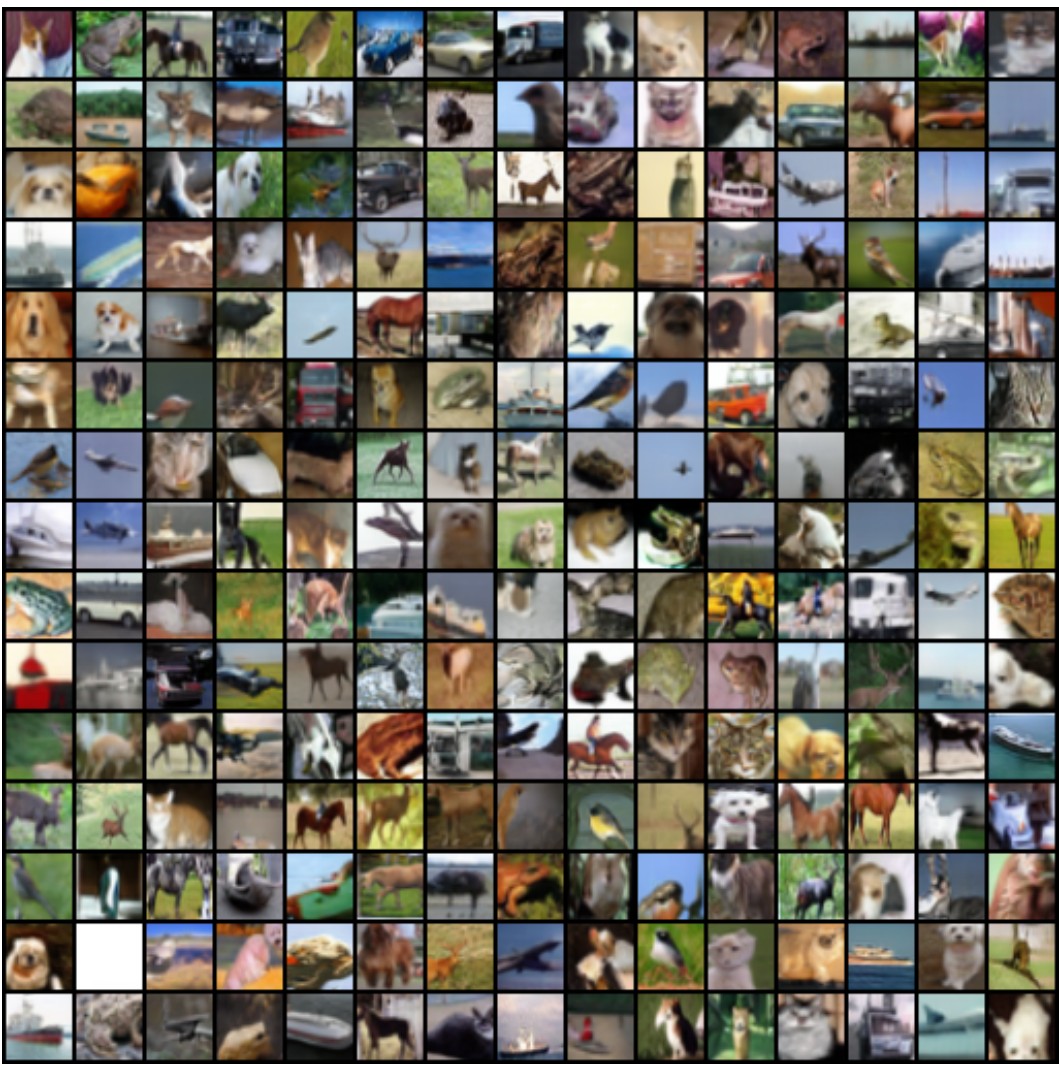

Figure 5: Randomly generated images from GAN trained with $\mathcal{L}_{\text{Gaussian}}$ only

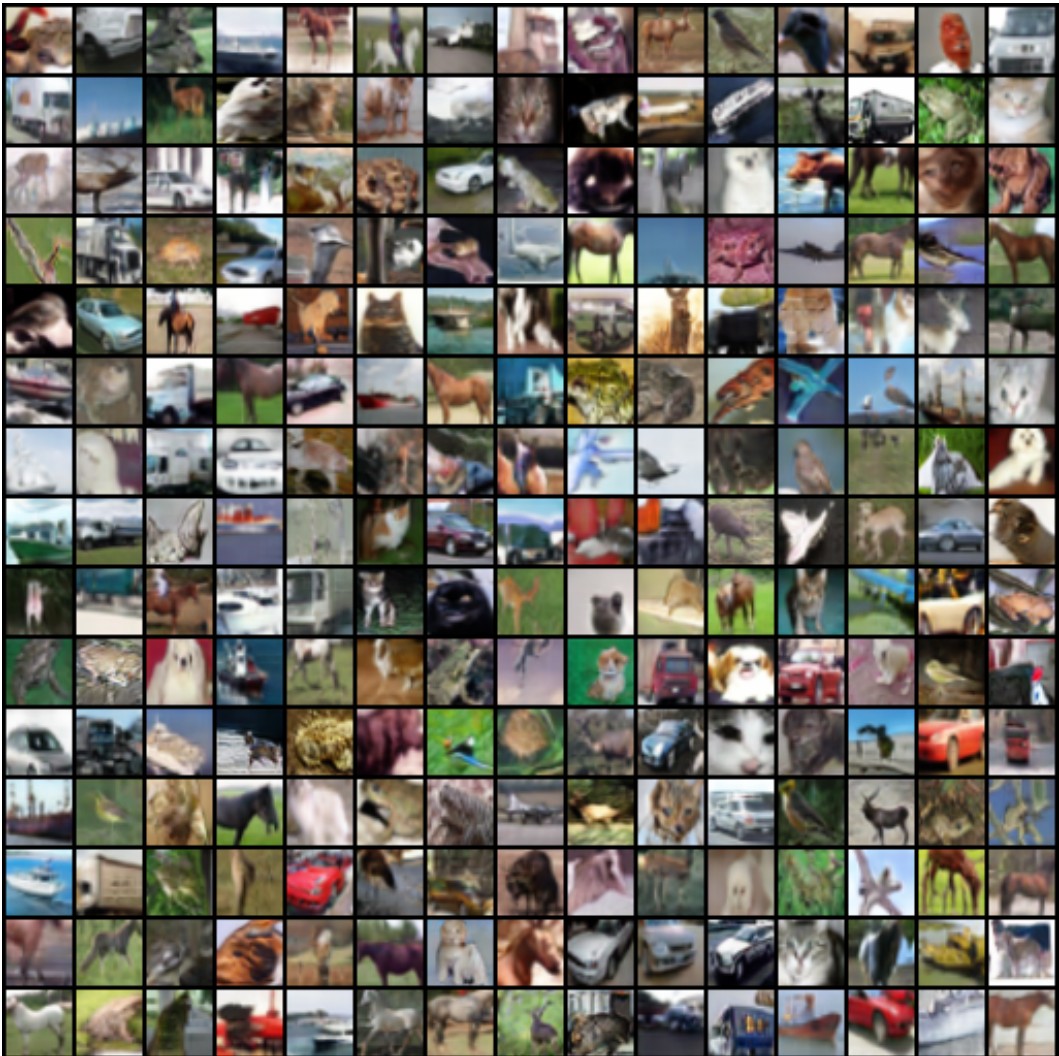

Figure 6: Randomly generated images from GAN trained with hinge loss

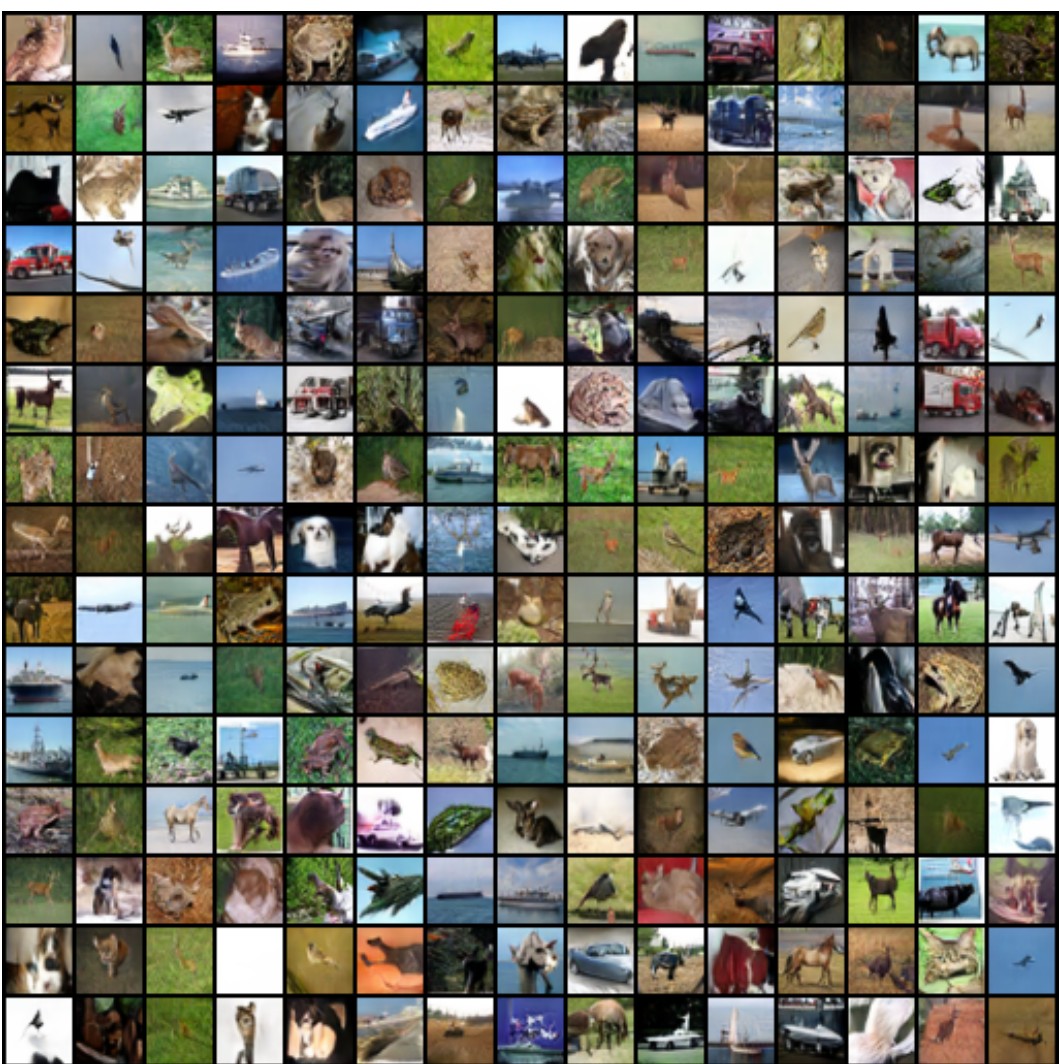

Figure 7: Randomly generated images from unconditional BigGAN

