# OpenReview forum: "Structural Adversarial Objectives for Self-Supervised Representation Learning"
_ICLR.cc/2023/Conference — Submitted to ICLR 2023_

### Official Review · Reviewer_GyrJ · 2022-10-23

**Confidence:** 3
**Correctness:** 3
**Technical Novelty And Significance:** 4
**Empirical Novelty And Significance:** 3
**Recommendation:** 5

**Clarity, Quality, Novelty And Reproducibility:**

The work is novel in multiple ways, the delivery is quite clear but it leaves out information hurting both understanding and reproducibility.


**Strength And Weaknesses:**

Strengths:
- Representation learning is a timely topic. Tackling it with generative models has been a promising approach but their usefulness had yet to be proven compared to their contrastive counterparts. The current submission is a big step in that direction.
- The approach is original in significant ways. It focuses on the discriminator, which is not usually exploited in GAN literature, and manages to use it as a feature learner.
- It also proposes a new GAN loss and a new GAN regularization scheme, which both seem useful for training GANs.

Weaknesses:
- The presentation of the technical parts could be made clearer:
  - In section 3.2, it seems that $f$ represents alternatively the individual outputs of $D$ as well as their distribution. It could be useful to clarify the notation. It can be especially confusing since the original adversarial objective, i.e. the binary classification objective between fake and real samples, can sometimes be presented as a JSD minimization of the distributions.
  - It could be useful to remind the reader of how VJP and JVP can be computed in the supplementary. I for one would welcome it.
  - Table 2 contains important information as it serves as an ablation study regarding the capacity of GAN discriminators to learn features. I don't think this aspect is discussed anywhere in the paper. Also, it isn't mentioned how the representations for the baselines in Table 2 have been extracted.
  - Have the contrastive baselines been trained on the datasets they are being evaluated on, like the GAN would be? Or have they been trained on ImageNet and evaluated only on those datasets, as is often the case in contrastive benchmarks? This information is important to put the results in context and should be made explicit.
- The paper should compare to other generative representation learning methods such as those they cite in their related work, or a more recent one [1].
- The authors might also want to discuss how their objectives relate to and differ from Feature Matching [2]. Here are some more recent examples of where it has been used [3,4].
- The BigGAN backbone is quite generic and useful, but somewhat outdated, especially for unconditional generation. Have the authors considered StyleGAN-2 [5] or FastGAN [6] backbones? Can the author discuss the potential issues of switching to such backbones?

---

[1] Ali Jahanian, Xavier Puig, Yonglong Tian, Phillip Isola. Generative Models as a Data Source for Multiview Representation Learning. ICLR 2022.

[2] Tim Salimans, Ian J. Goodfellow, Wojciech Zaremba, Vicki Cheung, Alec Radford, Xi Chen. Improved Techniques for Training GANs. NeurIPS 2016.

[3] Ting-Chun Wang, Ming-Yu Liu, Jun-Yan Zhu, Andrew Tao, Jan Kautz, Bryan Catanzaro. High-Resolution Image Synthesis and Semantic Manipulation With Conditional GANs. CVPR 2018.

[4] Liming Jiang, Changxu Zhang, Mingyang Huang, Chunxiao Liu, Jianping Shi, Chen Change Loy. TSIT: A Simple and Versatile Framework for Image-to-Image Translation. ECCV 2020.

[5] Tero Karras, Samuli Laine, Miika Aittala, Janne Hellsten, Jaakko Lehtinen, Timo Aila. Analyzing and Improving the Image Quality of StyleGAN. CVPR 2020.

[6] Bingchen Liu, Yizhe Zhu, Kunpeng Song, Ahmed Elgammal. Towards Faster and Stabilized GAN Training for High-fidelity Few-shot Image Synthesis. ICLR 2021.


**Summary Of The Paper:**

The paper tackles the problem of representation learning with GANs. To do so, it introduces a novel adversarial objective that is composed of a distribution-matching objective and a clustering objective. In addition, they propose a novel regularization algorithm to ensure the smoothness of the discriminator. Compared to the widespread spectral normalization, the proposed regularization does not control the spectral norm at each layer individually, allowing the discriminator to retain more capacity. The method is evaluated against contrastive state-of-the-art, on CIFAR-10, CIFAR-100, and ImageNet-10.

**Summary Of The Review:**

The work has relevant contributions for both GANs in general as well as their application to representation learning.
It tackles the problem in a novel and interesting way.
However, too many details are missing, as well as comparisons, and some related work.

---

> ### Author Response · Authors · 2022-11-19
> **Response to Reviewer GyrJ**
>
> Thank you for your feedback.
>
> **For clarification:**
>
> > $\mathbf{1.}$  Notion of $\boldsymbol{f}$
>
> $\boldsymbol{f}$ indicates individual output from *D* and, at the same time, we assume $\boldsymbol{f}$ to be a sample from the Gaussian distribution, i.e., $\boldsymbol{f} \sim \mathcal{N}(\mathbf{\mu}\_{\boldsymbol{f}}, \mathbf{\Sigma}\_{\boldsymbol{f}})$. So we consistently use $\boldsymbol{f}$ to denote individual output.
>
> > $\mathbf{2.}$  Connecting vanilla GAN objectives to distance-based objectives
>
> We interpret the right-hand side of Equation 1, i.e., the binary classification on actual and generated samples, as a distance measurement between two distributions.  Then it reaches the definition of Equation 2 using distance *d*.  Then we instantiate such distance measurement with JSD.  We will rephrase our description of this section in the final version to clarify.
>
> > $\mathbf{3.}$  Introduction to VJP and JVP.
>
> VJP standards for Vector-Jacobian Product. It is often used as a fundamental computation block during the gradient backpropagation, where the gradient input is the vector, and the derivative of output w.r.t input is Jacobian. Because of the chain rule, we can efficiently compute VJP of a large model by chaining VJP over a sequence of layers.  We use this subroutine for the efficient implementation of the power iteration as it include multiplication between vector and the Jacobian of the model.  JVP works similarly but switches the computing order of Jacobian and Vector.  We will include this description in our final draft.
>
> > $\mathbf{4.}$  Details of Table. 2
>
> In Table 2, the BigGAN baseline uses its default discriminator, and the rest use the modified ResNet-18 as the discriminator.  Compared to the BigGAN, the second row, hinge loss, uses a larger discriminator and generator and performs better.  To extract representations from the discriminator, we follow the same standard across all methods: we apply average pooling on the last feature map of the backbone to produce a single vector, i.e., the image representation.
>
> > $\mathbf{5.}$ Dataset for training and evaluation.
>
> We do not transfer knowledge across datasets, and all methods are trained and evaluated on the same datasets, i.e., for the experiment on CIFAR-10, we only train our model on the CIFAR-10 training split and evaluate it on the test split.
>
> > $\mathbf{6.}$  Comparison to GenRep
>
> We compare against GenRep (Jahanian et al., 2021) by running an experiment on ImageNet-100 and report the results in the general reply.  We adopt a simper feature extractor (ResNet-18 vs. ResNet-50), yet outperform the compared methods on the linear probing metric (59.9\% vs. 55.0\%).
>
> > $\mathbf{7.}$  Ablation experiments with Feature matching and FastGAN.
>
> As suggested, we run ablation experiments by adopting Feature matching as an extra loss for the generator or switching the generator architecture from BigGAN to FastGAN.  We provide implementation details and report the results in the general reply.  We do not see substantial performance differences with these modifications.
>
> [1] Jahanian, A., Puig, X., Tian, Y., Isola, P. (2021, September). Generative Models as a Data Source for Multiview Representation Learning. In International Conference on Learning Representations.

---

> > ### Comment · Reviewer_GyrJ · 2022-12-03
> > **Response to the authors**
> >
> > Thank you for your clarifications. I will first go through each item and then a more global assessment.
> >
> > > 1. So we consistently use $f$ to denote individual output.
> > > 2. We will rephrase our description of this section in the final version to clarify.
> >
> > It doesn't seem to be the case in equation 2, 4 and 5. But with the clarification in point 2. it should be fine.
> > Thank you.
> >
> > > 3. VJP standards for Vector-Jacobian Product.
> >
> > Thanks for taking the time to detail it. After looking it up, it seems to be fairly standard terminology that I just wasn't aware of. I apologize for assuming it had an obscure concept tied to it. My bad, I retract my request to include details in the paper if the authors don't feel it is necessary.
> >
> > > 4. we follow the same standard across all methods: we apply average pooling on the last feature map of the backbone to produce a single vector, i.e., the image representation.
> >
> > This methodology should probably be included in the paper.
> >
> > > 5. We do not transfer knowledge across datasets, and all methods are trained and evaluated on the same datasets,
> >
> > This is a fairly important point because self-supervised methods tends to performs much better with more data. Training and evaluating on CIFAR and ImageNet-10 is unexpected for state-of-the-art contrastive methods.
> >
> > > 6. We adopt a simper feature extractor (ResNet-18 vs. ResNet-50), yet outperform the compared methods on the linear probing metric (59.9% vs. 55.0%).
> >
> > There might be an argument to be made for comparison with Tz only, but if you are claiming to be competitive with state-of-the-art methods, then you ought to present the Tx+Tz results in this situation.
> >
> > >  7.
> >
> > Thank you for the additional results. I should precise that I have not suggested experiments with Feature Matching, but merely pointed out that it shares some similarities with your method that could be worth discussing. Also, in the original formulation, Feature Matching was not proposed as an additional loss but as a replacement for the GAN objective for the generator.
> >
> > > Overall
> >
> > As stated in my review, I do believe the submission to be original and interesting in many ways. However, given the answers provided by the authors, I find that the evaluation methodology is not accurately described in the paper. Importantly, claiming that the proposed method is competitive with state-of-the-art contrastive methods without any precision is misleading as the methods are not compared on standard contrastive benchmarks, nor against the strongest results in Jahanian et al., 2021. I would advise the authors to be more thorough and precise in the description of their experimental setups and revise their claims.

---

> > > ### Author Response · Authors · 2022-12-03
> > > **Response to Reviewer GyrJ**
> > >
> > > The current dominance of view-consistent-based contrastive learning approaches results from a sequence of papers contributing to advances in both core approaches and engineering.  In this sequence, pioneering works proposed and validated new ideas on smaller datasets, while subsequent engineering efforts developed techniques to scale them up.
> > >
> > > For contrastive self-supervised learning, a concrete example of this evolution is from: (Wu et al., 2018) [1], which introduced the idea of an instance-level contrastive objective and demonstrated results on ImageNet, to: (He et al., 2021) [2], which developed a way to efficiently scale up implementation of that objective to datasets of one billion images.
> > >
> > > Our work is at the stage of a new idea for GAN-based representation learning, with results demonstrating a dramatic leap on CIFAR-10/100 and ImageNet-10.  Note that some results on ImageNet-100 are in our general reply above.  Demanding everything at once (i.e., a new idea must not only match competitors at similar scale, but best years of additional engineering effort) is a surefire way to impede the progress of science and avoid any risk of accepting interesting papers.
> > >
> > > For clarity, in our final draft, we will position work as achieving competitive performance on CIFAR-10/100 and ImageNet-10 in both the abstract and introduction section.
> > >
> > > [1] Zhirong Wu, Yuanjun Xiong, Stella X. Yu, and Dahua Lin.  Unsupervised Feature Learning via Non-Parametric Instance-level Discrimination.  CVPR, 2018.
> > >
> > > [2] Kaiming He, Haoqi Fan, Yuxin Wu, Saining Xie, and Ross Girshick.  Momentum Contrast for Unsupervised Visual Representation Learning.  CVPR, 2021.

---

> > > > ### Comment · Reviewer_GyrJ · 2022-12-03
> > > > **Response to the authors**
> > > >
> > > > I agree with you that asking for a generative model to beat current state-of-the-art on say MS-COCO is not realistic.
> > > > I am only asking that the claims are aligned with the provided evidence. This should not be a controversial demand.
> > > >
> > > > Now, would you care to elaborate about why the result reported in the rebuttal for Jahanian et al., 2021 does not include the full version of their approach, and why you are have not considered evaluating the IC-GAN baseline?
> > > > I personally do not think that your method necessarily needs to beat those two baselines, but it is definitely necessary that your method is properly positioned w.r.t those. Ideally, their scores should to be discussed in the paper.
> > > > Also, please understand that you not mentioning those aspects in your rebuttal at all is raising legitimate concerns.

---

> > > > > ### Author Response · Authors · 2022-12-04
> > > > > **Response to Reviewer GyrJ**
> > > > >
> > > > > > This should not be a controversial demand.
> > > > >
> > > > > We agreed in the previous reply.  As stated in the last sentence of our previous reply, “we will position work as achieving competitive performance on CIFAR-10/100 and ImageNet-10 in both the abstract and introduction section.”
> > > > >
> > > > > > Evaluation of GenRep (Jahanian et al., 2021)
> > > > >
> > > > > GenRep assumes access to a trained generative model; that model is one of the inputs to their method.  We are, from scratch, constructing a generative model and representation learner simultaneously.  Even before experimental comparison, there is a qualitative difference between their multi-step training approach and our single-step training approach.
> > > > >
> > > > > GenRep optimizes SimCLR objectives by generating positive pairs from: (1) decoding adjacent samples in the GAN's latent space, termed as Tz, and (2) running SimCLR-style augmentation on generated images, termed as Tx.  One of our aims is to eliminate the need for domain-specific view-consistency objectives.  GenRep with Tx includes view-consistency objectives, placing it outside this setting.  We chose GenRep on Tz as an appropriate baseline for comparison, as neither it nor our method depend on domain-specific data augmentation.  In the general reply, we show a clear advantage (59.9 vs. 55.0) over this baseline, and we even use a simpler feature extractor, ResNet18, rather than the ResNet50 of GenRep.
> > > > >
> > > > > It is worth pointing out that, from Table 1 and Table 2 of GenRep, their best setting, Tx + Tz, does not beat SimCLR, which is a fair comparison of two methods relying on view-consistency objectives.  We do compare with SimCLR and we are open to also including, in our final version, a comparison with the GenRep Tx + Tz variant.  This comparison would be between systems using different data augmentation regimes, but would provide additional context.
> > > > >
> > > > > > Comparison to IC-GAN.
> > > > >
> > > > > To collect the nearest neighbors for use as their conditional input, IC-GAN uses a pre-trained feature extractor (SWAV for unlabeled settings and supervised ResNet50 pre-trained on ImageNet for class-conditional settings).  They freeze this feature extractor during the course of training.  Regardless of whether their learned discriminator improves over SWAV, as input to their system, IC-GAN requires a feature extractor to be provided.  That is, IC-GAN assumes access to a trained representation learning system -- but that is precisely the problem we are trying to solve: how to train a representation learner from scratch.
> > > > >
> > > > > One might imagine running IC-GAN with our system serving as the original feature extractor and comparing it with IC-GAN on SWAV, but this is a strange kind of secondary comparison.  It is SWAV, or SimCLR, or other representation learning systems that are the primary and correct choice of component for direct comparison.
> > > > >
> > > > > Finally, as also explained in our most recent response to Reviewer 9r54, IC-GAN itself focuses entirely on improving image generation quality and makes no claims with respect to representation learning.  There is not a single experimental result in the IC-GAN paper that evaluates IC-GAN as a representation learner.  Every IC-GAN result table measures perceptual quality of generated images (FID, IS, and LPIPS metrics); there is no evaluation of latent representations for downstream tasks.

---

> > > > > > ### Comment · Reviewer_GyrJ · 2022-12-06
> > > > > > **Response to the Authors**
> > > > > >
> > > > > > Thank you for the clarifications and motivating your choices. The arguments are noted.
> > > > > > Other questions I wanted to ask after thinking about the other reviews is about the ablation study in Table 2.
> > > > > >
> > > > > > 1) One important difference between the Gaussian Loss and the baselines is that the Gaussian loss acts directly on $f$ and do use a learnt linear layer to interpret $f$ as a Loss. This could be an explanation about why $f$ is less noisy and better suited for t-SNE visualisation and other downstream tasks regardless of the actual properties of the Gaussian Loss.
> > > > > > Do you have any additional insights on the effect of having a GAN objective on a representation vector directly? That would allow to better disentangle the effects of the Gaussian Loss.
> > > > > >
> > > > > > 2) Apparently, with the Gaussian Loss, the clustering loss has small effects on the representation learning part, but huge impact on FID.
> > > > > > Have you considered performing the ablation of adding the clustering loss to BigGAN or HingeLoss baseline to see if the same findings occurs?

---

> > > > > > > ### Author Response · Authors · 2022-12-09
> > > > > > > **Response to Reviewer GyrJ**
> > > > > > >
> > > > > > > Thanks for the comments.
> > > > > > >
> > > > > > > 1.One benefit is that we can directly optimize for proper structures of f, e.g., covariance, to enhance its role as representation. Another advantage is we do not have to worry about potential asynchronized updating between f and the last linear layer.
> > > > > > >
> > > > > > >
> > > > > > > 2.One explanation for the boost in FID is that: our clustering loss is instance-wise grouping loss, and it encourages generated images to be similar to their adjacent real examples.
> > > > > > >
> > > > > > > Thanks for suggesting this ablation experiment. We follow your suggestion and run an experiment on CIFAR-10 by adding grouping loss to the Hinge Loss baseline. We use the final output of the ResNet backbone, a 512-dimensional vector followed by l2 normalization, as the feature vector $f_i, f^g_i$ and $f_{i,j}, f^g_{i,j}$ in $\mathcal{L}\_{group}$,  $\mathcal{L}_{adv group}$. Due to the time limit, we only report a partial training pass at 700 epochs and compare it with the original hinge loss at the same epoch.
> > > > > > >
> > > > > > > | Method | KMeans | SVM | 1/5 KNN | IS | FID |
> > > > > > > | :---: | :----: | :---: | :----: | :----: | :----: |
> > > > > > > | Hinge Loss | **39.51** | **77.80** | 71.09/74.12 |**8.13**|**22.04**|
> > > > > > > | Hinge Loss + $\mathcal{L}_{group}$ | 17.99 | 76.50| **73.91**/**76.14**| 8.07 | 24.57|
> > > > > > >
> > > > > > > Interestingly, the results of Hinge Loss at 700 epochs are slightly better than that in 1000 epochs, possibly due to the commonly observed mode collapsing in the vanilla GAN model.
> > > > > > > Extra grouping loss optimizes neighbor structures and improves KNN accuracy compared to the Hinge Loss baseline. Because we do not optimize distribution structures as in Gaussian loss, grouping adjacent embedding may aggravate feature collapse and yield worse KMeans, image quality, and SVM accuracy.

---

### Official Review · Reviewer_9r54 · 2022-10-24

**Confidence:** 4
**Clarity, Quality, Novelty And Reproducibility:** 1. The paper is clear, well-written, …
**Correctness:** 3
**Technical Novelty And Significance:** 3
**Empirical Novelty And Significance:** 3
**Recommendation:** 6

**Strength And Weaknesses:**

Strengths:
1. The paper is well-written and easy to follow.
2. The paper is well-motivated, and it is technically and theoretically sound.
3. The proposed model outperforms BigGAN in terms of image generation quality.
4. The idea of self-supervised feature representation learning using a discriminator is interesting and novel to the best of my knowledge.

Weaknesses:
1. There are some grammatical errors in the text, and some sentences are too long to follow. E.g.: Page 1, Yet, to improve...
2. There is no discussion on different discriminator architectures, such as the patchGAN discriminator or feature extractors as discriminators [1,2]. Also, a more detailed comparison against ICGAN can improve the paper.

[1] Sungatullina, Diana, et al. "Image manipulation with perceptual discriminators." ECCV 2018.

[2] Mao, Xin, et al. "Is discriminator a good feature extractor?." arXiv 2019.

3. The evaluation setting is not common in self-supervised approaches. Usually, linear classification and kNN classifiers are used. In this paper, they use SVM and Kmeans.

4. The computational complexities of the networks in Tab. 1 are not compared.

5. The proposed model only outperforms previous work on one metric.

Minor:
The references for related work in Tab. 1 could be added.

**Summary Of The Paper:**

This work proposes to use the discriminator in a GAN as a feature extractor for self-supervised representation learning. Assuming that both real and fake features from the discriminator follow a gaussian distribution, the authors propose a loss based on the distance between the real and fake gaussian distributions. They also propose performing clustering on the extracted features and use this as an extra loss term for the discriminator optimization. Their proposed method outperforms SOTA self-supervised learning approaches on small-scale datasets and using the K-means metric.

**Summary Of The Review:**

The paper has both strengths and weaknesses. I have some concerns regarding the evaluation setting and discussion on existing literature.

---

> ### Author Response · Authors · 2022-11-19
> **Response to Reviewer 9r54**
>
> Thank you for your feedback.
>
> **To your concerns:**
>
> > $\mathbf{1.}$  Grammatical error
>
> Thank you for pointing that out.  We will revise the text for the final version.
>
> > $\mathbf{2.}$  Comparison to other GAN variants.
>
> Thank you for the suggested references.
>
> Using a pre-existing feature extractor as a discriminator defeats our objective, since we focus on representation learning rather than improving image quality.  If we already had a good feature extractor available, we would not need to do anything.  Our goal is to, in an unsupervised manner, build a discriminator that also subsequently serves as a feature extractor for downstream tasks.
>
> PatchGAN  (Isola et al., 2017) implements a discriminator using an image patch as input, which is generally useful for conditional scene image generation rather than unconditional generation tasks.
>
> ICGAN (Casanova et al., 2021) optimizers image generation with nearest neighbor samples collected from a pre-trained encoder.  The major difference between our method and ICGAN is that our method uses the learned representation to optimize clustering objectives, while the encoder for ICGAN is fixed and pre-trained.
>
> In our final version, we will discuss these references [1,2] in related work.
>
> > $\mathbf{3.}$  Evaluation with linear classification and KNN classifier.
>
> We provide the result on the suggested metric in Tables 1 and 2 of the general reply.
>
> > $\mathbf{4.}$  Model's computation complexity
>
> Here is a reference for computation complexity and model parameters, which we will add to the final paper:
>
> | Method                | Paramters    | FLOPs  | Dataset |
> | :---:                 |    :----:    | :----: | :----: |
> | Contrastive Encoder   | 11.17M       | 1.42G  | CIFAR-10/100 |
> | Our Discriminator     | 11.50M       | 1.10G  | CIFAR-10/100 |
> | Our Generator         | 4.87M        | 1.36G  | CIFAR-10/100 |
> |--|--|--|--|
> | Contrastive Encoder   | 11.70M        | 2.44G  | ImageNet-10 |
> | Our Discriminator     | 11.70M       | 2.44G  | ImageNet-10 |
> | Our Generator         | 70.19M       | 41.24G | ImageNet-10 |
>
> > $\mathbf{5.}$  Results compared to contrastive methods.
>
> The proper interpretation of Table 1 (page 8 in our manuscript) is that it demonstrates our method can achieve compatible performance to the current state-of-the-art method, without leveraging a hand-crafted view-consistent data augmentation regime.  In the general reply, we further showcase the sensitivity of contrastive methods under varying data augmentation schemes and demonstrate our method's stability across those cases.
>
> [1] Isola, P., Zhu, J. Y., Zhou, T., Efros, A. A. (2017). Image-to-image translation with conditional adversarial networks. In Proceedings of the IEEE conference on computer vision and pattern recognition (pp. 1125-1134).
>
> [2] Casanova, A., Careil, M., Verbeek, J., Drozdzal, M., Romero Soriano, A. (2021). Instance-conditioned GAN. Advances in Neural Information Processing Systems, 34, 27517-27529.

---

### Official Review · Reviewer_XWaT · 2022-10-25

**Confidence:** 4
**Clarity, Quality, Novelty And Reproducibility:** 1. The idea that tasks the discrimina…
**Correctness:** 2
**Technical Novelty And Significance:** 2
**Empirical Novelty And Significance:** 2
**Recommendation:** 3

**Strength And Weaknesses:**

The proposed structural adversarial objectives are interesting and seem novel. The performance of the proposed GAN especially on K-mean clustering is promising. However, I have the following concerns:
1. Actually, this is a kind of work that utilizes a generative model together with clustering for SS representation learning. Therefore, deep generative clustering, a.k.a., generative model (GAN/VAE) with clustering, e.g., [1,2,3,4], is closely related to this work, which is missed in the Related Work part of this paper.
2. I think the proposed work can be better positioned in the literature if the motivation is to target deep generative clustering instead of SS representation learning. I would expect a detailed analysis about the superior performance of the proposed generative clustering to the existing generative clustering works.
3. [1] also uses discriminator features for clustering. From this aspect, the idea claimed in this work that tasks the discriminator with additional structural modeling is not novel.
4. The experimental study is not totally consistent with the original motivation. First, augmentation is adopted though simple. Second, the experiments can be more convincing if conducted on different modalities of data, like text. It can verify the generality of the proposed method and also demonstrate the data augmentation should be specified in terms of data domains.
5. Gaussian assumption encouraging smoothness and diversified representation is not well justified.


Some questions:
1. Why spectral normalization used in Miyato et al. 2018 will harm model capacity but the spectral normalization used in this work will not?
2. How to decide which layers of D as clustering features?
3. What is the size of the maintained memory bank $f_m$?
4. For coarse-scale optimization with Gaussian, is it using mini-batches to determine the covariance and mean? If so, is it problematic since Eq. (4)/(5) is supposed to be defined on the whole data?
5. Why “regularizing JD(x) on l2 normalized embedding f will enlarge the norm of f˜ throughout training and eventually destabilizes the system; while regularizing JD(x) on f˜operates oppositely”?


Errors
1. In Section 2.2, “and demonstrates their sensitivity to the parameters of augmentation schemes.”-→demonstrate
2. In Eq. (2), JSD should equal to $1/2*D_{KL}$…
3. In $L_{advgroup}$, $f_i^g$ is supposed to be $f_i$?
4. $\\{f_{i,j}\\}_{j=1}^{k}$, $k$ is supposed to be $K$.
5. In Alg 1, $s$ should be $S$.
6. Fig. 2, (4) should be (d).

References

[1] Liu, S., Wang, T., Bau, D., Zhu, J. Y., & Torralba, A. (2020). Diverse image generation via self-conditioned gans. In Proceedings of the IEEE/CVF conference on computer vision and pattern recognition (pp. 14286-14295).

[2] Noroozi, M. (2020). Self-labeled conditional gans. arXiv preprint arXiv:2012.02162.

[3] Guo, X., Gao, L., Liu, X., & Yin, J. (2017, August). Improved deep embedded clustering with local structure preservation. In Ijcai (pp. 1753-1759).

[4] Jiang, Z., Zheng, Y., Tan, H., Tang, B., & Zhou, H. (2017, January). Variational Deep Embedding: An Unsupervised and Generative Approach to Clustering. In IJCAI.


**Summary Of The Paper:**

This paper proposed a GAN variant with structural adversarial objectives for self-supervised (SS) representation learning, which aims to achieve a general SS representation learning, especially escaping dependence upon the hand-crafted elements guiding data augmentation or proxy task design. In particular, at a coarse scale, a JSD divergence between the discriminator features of real samples and generated samples is defined as an adversarial objective for GAN.  At a finer scale, an adversarial clustering objective is defined, which groups adjacent real embeddings (features extracted by the discriminator) to form a cluster and adversarially attracts the generated embedding towards the nearby cluster center. Furthermore, an efficient regularization scheme that approximates the spectral norm of the Jacobian is introduced to regularize the discriminator’s smoothness. Experiments demonstrate their proposed GAN achieves results that compete with networks trained by state-of-the-art contrastive approaches.

**Summary Of The Review:**

The proposed structural adversarial objectives are interesting and seem novel. The performance of the proposed GAN especially on K-mean clustering is promising. However, the authors do not well position their work and miss the comparison with closely related works. Thus, the contributions of this work cannot be clearly demonstrated. In addition, the empirical support is not consistent with the original motivation. Last, the clarity of this work needs improvement.

---

> ### Author Response · Authors · 2022-11-19
> **Response to Reviewer XWaT (1/2)**
>
> Thank you for the detailed comments.  Please also see our reply to all reviewers, explaining positioning of the work and providing additional experimental results.
>
> **To your concerns:**
>
> > $\mathbf{1.} \\&  \mathbf{2.}$  Comparison to generative clustering approaches.
>
> We disagree with the comment that our work "Utilizes a generative model together with clustering for SS representation learning".
>
> As stated in the general reply, our main objective is representation learning rather than clustering.  Clustering is one of the metrics used to evaluate our learned representation.  In our learning pipeline, the clustering loss is an optional auxiliary objective to further reshape the embedding space of the discriminator in favor of downstream classification tasks.  As shown in Table 2 (rows 3,4) and Figure 3 (c), our learned embedding already exhibits decent category-wise structures without clustering objectives.
>
> > $\mathbf{3.}$  Compared to Self-conditional GAN (Liu et al., 2020a).
>
> Self-conditional GAN (Liu et al., 2020a) is motivated by producing diversified images rather than learning feature representations.  They leverage the discriminator's features to cluster images.  However, they do not have any training objectives to encourage the discriminator to produce better representations than a vanilla GAN.  As presented in the general reply, our method significantly outperforms self-conditional GAN in terms of clustering performance.
>
> > $\mathbf{4.}$  Data augmentation in training pipeline.
>
> Our motivation is to learn representations without hand-crafted view-invariant objectives (the purpose of augmentation in contrastive approaches).  The primary difficulty of extending contrastive learning for different tasks is the need to re-design augmentation for each task to achieve a sweet spot, which maximizes task-wise mutual information while minimizing mutual information on augmented samples, as illustrated in Figure 1 of InfoMin (Tian et al., 2020).
>
> However, augmentation in our pipelines is just for improving the generation quality, and our method relies less on data augmentation.  To demonstrate that, in the general reply, we show our method outperforms contrastive methods when using the same data augmentation and even learns without any data augmentation; contrastive approaches only work well with a full spectrum of augmentation.
>
> > $\mathbf{5.}$  Smoothing and diversified representation with Gaussian assumption.
>
> Optimizing Gaussian objectives in D's round aims to maximize the total covariance of l2 normalized feature vectors, which amounts to maximizing the pairwise distance between samples and yielding non-collapsed representations. From the definition of the Lipschitz constant, maximizing pairwise feature distance amounts to minimizing the Lipschitz constant, which implies a smoother mapping to the embedding space.  We will revise our final draft to clarify these notions.

---

> > ### Author Response · Authors · 2022-11-19
> > **Response to Reviewer XWaT (2/2)**
> >
> > **To your questions:**
> >
> > > $\mathbf{1.}$  Influence of regularization on the capacity of discriminator
> >
> > Spectral Norm normalizes parameter $\mathbf{W}$ by its spectral norm $\sigma(\mathbf{W})$ to enforce 1-Lipschitz for each layer. However, it usually ends up over-minimizing the  $||D|| \_{Lip}$ due to the chain rule of Lipschitz constant over layers, i.e., $||D|| \_{Lip}\leq \prod\_{l=1}^{L}{\sigma(\mathbf{W}\_l)}$. For example, as shown in Figure 3 of GradNorm (Wu et al., 2021), $||D||\_{Lip} \approx 0$ for a 9-layer discriminator.
> >
> > Our proposed regularization instead optimizes the model-wise singular value allowing intermediate layers to adjust their layer-wise Lipschitz constant and sustain capacity.  For example, after convergence, the averaged $||J_D(x)||\_2$, estimated over 256 images, is 2.01. However, the averaged layer-wise spectral norm for 4 ResNet blocks are 2.14, 2.68, 4.67 and 8.33 respectively.
> >
> > > $\mathbf{2.}$  Which layers of D are used for representation?
> >
> > All proposed objectives, including our clustering targets, are applied to the final output of the discriminator, i.e., $\boldsymbol{\hat{f}}$, $\boldsymbol{\hat{f}^g}$
> >
> > > $\mathbf{3.}$  Size of memory bank $\boldsymbol{f}\^m$
> >
> > We set $|\boldsymbol{f}\^m| = 10,240$. We intentionally limit the memory bank size to smaller than the training set so that searching for the nearest neighbor doesn't pick the augmentation from the same image.
> >
> > > $\mathbf{4.}$  Global covariance approximation with minibatch samples
> >
> > The covariance is defined over the mini-batch sample.  With a sufficient batch size, which is $512$ in our experiment, the mini-batch statistic provides a reasonable approximation over the global distribution statistic.
> >
> > > $\mathbf{5.}$  Choices of $\boldsymbol{J}\_D(\mathbf{x})$ on $\boldsymbol{f}$ or $\hat{\boldsymbol{f}}$ and its impact on training stability
> >
> > Suppose we have loss $\mathcal{L}$ operated on l2 normalized feature $\hat{\boldsymbol{f}} := \frac{\boldsymbol{f}}{||\boldsymbol{f}||\_2}$,
> > we can show $||\boldsymbol{f}||\_2$ is always increased after gradient updating because $||\boldsymbol{f} + \lambda \partial \mathcal{L} / \partial \boldsymbol{f}||\_2 \geq ||\boldsymbol{f}||\_2 $ for $\lambda \geq 0$. (For detailed derivation and analysis, please check Equation 4 and Figure 4 of SphereFace (Wang et al., 2017)). Therefore, optimizing our primary adversarial objectives, i.e. $\mathcal{L}\_{\rm{Gaussian}}$, $\mathcal{L}\_{\rm{Cluster}}$, will increase $||\boldsymbol{f}||\_2$ since they are operated on $\hat{\boldsymbol{f}}$.
> >
> > As we defined in Equation 6, the regularization loss $\mathcal{L}\_{\rm{reg}}$ includes $||\boldsymbol{J}\_D(x) - \rm{Lip}||_2$ term, and we can compute $\boldsymbol{J}\_D(x)$ on either $\boldsymbol{f}$ or $\hat{\boldsymbol{f}}$. If we compute $\boldsymbol{J}\_D(x)$ on $\boldsymbol{f}$, by definition of the Jacobian matrix, minimizing $||\boldsymbol{J}\_D(x)||\_2$ will reduce $||\boldsymbol{f}||\_2$. If we compute $\boldsymbol{J}\_D(x)$ on $\hat{\boldsymbol{f}}$, based on the analysis above on $\hat{\boldsymbol{f}}$, minimizing $||\boldsymbol{J}\_D(x)||\_2$ will increase $||\boldsymbol{f}||\_2$.
> >
> > We empirically find maintaining proper $||\boldsymbol{f}||\_2$ is important for maintaining the dynamical balance for adversarial training. Therefore we choose to compute $\boldsymbol{J}\_D(x)$ on $\boldsymbol{f}$ and add extra hinge loss to prevent $||\boldsymbol{f}||\_2$ from further decreasing.
> >
> >
> >
> > > $\mathbf{6.}$  Typos and errors
> >
> > Thank you for pointing out the typos; we will fix those in our final version.
> >
> > [1] Tian, Y., Sun, C., Poole, B., Krishnan, D., Schmid, C., Isola, P. (2020). What makes for good views for contrastive learning?. Advances in Neural Information Processing Systems, 33, 6827-6839.
> >
> > [2] Liu, S., Wang, T., Bau, D., Zhu, J. Y., Torralba, A. (2020a). Diverse image generation via self-conditioned gans. In Proceedings of the IEEE/CVF Conference on Computer Vision and Pattern Recognition (pp. 14286-14295).
> >
> > [3] Wang, F., Xiang, X., Cheng, J., Yuille, A. L. (2017, October). Normface: L2 hypersphere embedding for face verification. In Proceedings of the 25th ACM International Conference on Multimedia (pp. 1041-1049).
> >
> > [4] Wu, Y. L., Shuai, H. H., Tam, Z. R., Chiu, H. Y. (2021). Gradient normalization for generative adversarial networks. In Proceedings of the IEEE/CVF International Conference on Computer Vision (pp. 6373-6382)

---

> > > ### Comment · Reviewer_XWaT · 2022-11-23
> > > **The key component (unimodal Gaussian loss) beneficial to category-wise structures is counterintuitive but lacks deep analysis**
> > >
> > > I appreciate the authors’ efforts to address my concerns. However, I still have some concerns after reading all reviews and responses.
> > >
> > > 1. As clarified by the authors, their learned embedding already exhibits decent category-wise structures without clustering objectives. This is actually against my knowledge. I originally thought the category-wise structures are mainly induced by the clustering objective (this is why I suggest positioning this work on the topic of generative clustering). However, the k-means clustering accuracy even decreases after adding the clustering objective (Table 2). This is not normal. I expect an explanation for this.
> > >
> > > 2. On the other hand, I feel the unimodal Gaussian loss is wrong to learn the structured representations. I think a multi-modal Gaussian loss is more reasonable. Although the empirical results indeed show the efficacy of the unimodal Gaussian loss, it is counterintuitive. The authors did not give any deep analysis for this. I cannot understand how the unimodal Gaussian loss can induce decent category-wise structures.
> > >
> > > 3. Why this work can outperform other generative clustering works, like ClusterGAN, self-conditioned GAN in terms of category-wise structures? The answer to this question may also help deeply analyze this work.

---

> > > > ### Author Response · Authors · 2022-11-26
> > > > **Adversarial unimodal Gaussian loss suffices for learning categorical representation**
> > > >
> > > > Thanks for the discussion.
> > > >
> > > > > 1. representation learning and category-wise structures without clustering objectives
> > > >
> > > > Standard GAN models, without any clustering objectives, still learn useful and nontrivial representations.  Table 2 shows this result: the baseline methods BigGAN and Hinge Loss already yield decent performance for an SVM classifier trained on their representations.  Our proposed losses serve to further improve on these baselines.
> > > >
> > > > There are several discussions on the functionality of discriminators.  For example, (Che et al., 2020) shows that the discriminator in a GAN behaves as an energy-based model and learns the unnormalized image distribution.  Following this interpretation, one effect of maximizing the covariance determinant in our proposed Gaussian loss is to serve as a repulsion force, which has been shown to improve the performance of an energy-based model (Du et al., 2020) and yield highly non-uniform learned clustering structures in contrastive learning.
> > > >
> > > > Our clustering loss optimizes a mean-shift objective by grouping neighbor embeddings, rather than directly targeting a K-Means objective to produce K separated groups.  In our experiments, we find that the SVM performance is highly correlated with KNN accuracy.  Therefore, we adopt the mean-shift objective to optimize KNN features in order to further improve the representation; efficacy can be seen by the improvement of SVM performance in Table 2 (88.32 to 89.76).  As an evaluation metric, K-means clustering has high variance, so the gap between 80.55 and 80.11 in Table 2 is not significant enough to draw a conclusion.
> > > >
> > > > > 2. unimodal Gaussian loss is wrong to learn the structured representations
> > > >
> > > > Variational autoencoders (VAEs) are one counterexample.  For VAEs, a unimodal Gaussian is a popular and effective prior that leads to trained models which capture informative latent structures.  As shown in Figure 2 (A),(C) of (Makhzani et al., 2015), training VAEs and adversarial autoencoders (AAEs) with a unimodal Gaussian prior produces models that clearly encode category-wise structures in their latent space.
> > > >
> > > > At an intuitive level, a unimodal Gaussian prior serves to prevent collapse of the latent distribution (akin to uniform repulsion objectives in contrastive learning); it is not a hard constraint and does not prevent clusters from emerging.  Similarly, in contrastive learning, uniform repulsion is not perfectly satisfiable by the trained model; clusters emerge, and repulsion is partially satisfied (i.e., between clusters more so than within).
> > > >
> > > > > 3. clusterGAN, self-conditioned GAN
> > > >
> > > > From recent analysis of the self-supervised clustering approaches, the significant boost of clustering performance is primarily from employing better representation learning objectives.  Please check Table 2 of (Shen et al., 2021) for a comprehensive comparison.  In our Table 2, we also show that our Gaussian objectives alone suffice to boost clustering performance from 36.41 (hinge loss) to 70.76.
> > > >
> > > > On the other hand, producing cluster-wise structures, i.e., constraining the output space to produce separated groups, is insufficient for decent clustering performance.  Without proper representation learning targets to maintain embedding structures, e.g., nearest neighbor, the clustering result is usually inconsistent with the categorical distribution.
> > > >
> > > > [1] Che, T., Zhang, R., Sohl-Dickstein, J., Larochelle, H., Paull, L., Cao, Y., Bengio, Y. (2020). Your gan is secretly an energy-based model and you should use discriminator driven latent sampling. Advances in Neural Information Processing Systems, 33, 12275-12287.
> > > >
> > > > [2] Du, Y., Li, S., Tenenbaum, J., Mordatch, I. (2020). Improved contrastive divergence training of energy based models. arXiv preprint arXiv:2012.01316.
> > > >
> > > > [3] Makhzani, A., Shlens, J., Jaitly, N., Goodfellow, I., Frey, B. (2015). Adversarial autoencoders. arXiv preprint arXiv:1511.05644.
> > > >
> > > > [4] Shen, Y., Shen, Z., Wang, M., Qin, J., Torr, P., Shao, L. (2021). You never cluster alone. Advances in Neural Information Processing Systems, 34, 27734-27746.

---

> > > > > ### Comment · Reviewer_XWaT · 2022-12-06
> > > > > **Why the proposed Gaussian loss yielding better clustering structures is still not clear.**
> > > > >
> > > > > Thanks for the responses.
> > > > >
> > > > > 1. The authors claim that the representations of GANs are structured. Is this finding first proposed in this work? Does this phenomenon depend on specific network structures (e.g., BigGAN) or GANs loss (hinge loss)? It would be better if the paper analyzes their work beginning from this finding “the representations of GANs are structured”.
> > > > >
> > > > > 2. The authors claim that maximizing the covariance determinant in the proposed Gaussian loss helps yield highly non-uniform learned clustering structures in contrastive learning. This seems not been proven before. The repulsion does not guarantee categorical representation. I am still not clear why this loss suffices for learning categorical representation.
> > > > >
> > > > > 3. The authors did not give convincing reasons for why this work outperforms other generative clustering works.
> > > > > First, the adopted Gaussian loss in this work is quite different from those self-supervised representation learning losses, so those self-supervised clustering approaches cannot be used to support this work.
> > > > > Second, the adopted nearest neighbor objective is claimed as a proper representation learning target to maintain embedding structures. Those GANs with clustering themselves can boost good representations since they are learning data distribution (data manifold), also clarified by the authors (the first point).

---

> > > > > > ### Author Response · Authors · 2022-12-09
> > > > > > **Response to reviewer XWaT**
> > > > > >
> > > > > > We believe we answered many of the questions in our previous replies, but will elaborate below.
> > > > > >
> > > > > > >Q1: The authors claim that the representations of GANs are structured. Is this finding first proposed in this work? Does this phenomenon depend on specific network structures (e.g., BigGAN) or GANs loss (hinge loss)? It would be better if the paper analyzes their work beginning from this finding “the representations of GANs are structured”.
> > > > > >
> > > > > > A1: As mentioned in our introduction, Radford et al., 2015 is the first paper showing the discriminator learns a structural representation. See Figures 4 and 7 of Radford et al. for a visual demonstration of interpolation and vector arithmetic which reveals latent space structure. We analyze different generators (FastGAN) for comparison in the general reply and the ablated experiments on GAN losses in Table 2 of the paper. Please refer to those tables for details.
> > > > > >
> > > > > > >Q2: The authors claim that maximizing the covariance determinant in the proposed Gaussian loss helps yield highly non-uniform learned clustering structures in contrastive learning. This seems not been proven before. The repulsion does not guarantee categorical representation. I am still not clear why this loss suffices for learning categorical representation.
> > > > > >
> > > > > > A2: We only say the repulsion term helps improve the representation, which is precisely quoted in your comment, rather than guaranteeing categorical representation. As we mention in the previous reply and Table 2 shows, vanilla GAN suffices for learning categorical representation. We provide one potential explanation for this in the previous reply, commenting on the connection to energy-based models. In this research area, the results from both prior work and our work are empirical; there is intuition, but not proofs that follow from theory.
> > > > > >
> > > > > > >Q3: The authors did not give convincing reasons for why this work outperforms other generative clustering works. First, the adopted Gaussian loss in this work is quite different from those self-supervised representation learning losses, so those self-supervised clustering approaches cannot be used to support this work. Second, the adopted nearest neighbor objective is claimed as a proper representation learning target to maintain embedding structures. Those GANs with clustering themselves can boost good representations since they are learning data distribution (data manifold), also clarified by the authors (the first point).
> > > > > >
> > > > > > A3: Proof is in our empirical results. An intuitive explanation is as follows. Without proper feature learning objectives, the clustering objective alone is insufficient for learning categorical distribution. This phenomenon is universally observed in recent state-of-the-art unsupervised clustering work, and the comparison between our work and suggested generative clustering work also supports this claim. In Table 2, we show that our Gaussian loss and larger discriminator with proper regularization, rather than the clustering loss, primarily contributes to performance improvement on all metrics. So even for clustering metrics, representation learning objectives still provide the primary boost and the clustering objectives are mostly secondary.
> > > > > >
> > > > > > >Q4: I still hold my original opinion that this work can be better positioned in the literature if the motivation is to target deep generative clustering. And also, from this perspective, the efficacy of this work can be more deeply analyzed.
> > > > > >
> > > > > > A4: No. As emphasized throughout our submission, our primary aim is feature learning, not clustering. As shown in our previous reply, clustering is one of the metrics used to evaluate our learned representation, and clustering objectives serve as optional auxiliary objectives to further improve the representation. It is improper to position our work with clustering as the primary goal; representation learning is the primary goal.

---

### Author Response · Authors · 2022-11-19
**Response to all Reviewers (1/2)**

We thank the reviewers.  However, several comments mischaracterize our primary objectives and consequently underestimate the impact of our contribution: our method elevates GANs to being a competitor to the state-of-the-art self-supervised learning approaches.  We enhance the GAN discriminator to be a component that maps data to latent semantic representations, while maintaining the GAN generator's ability to produce realistic samples.  Representation learning, rather than clustering, is both our goal and the appropriate evaluation setting.

Our focus on turning the GAN discriminator into a competitive representation learner differs from that of much recent work on GANs.  We are reviving an older ambition of unsupervised learning using simple architectures with paired components mapping data to latent vectors and vice-versa.  Autoencoders are a classic example, while BiGANs (Donahue et al., 2016) are a modern example in the GAN framework, though neither are competitive with state-of-the-art self-supervised learning.  We do achieve competitive performance to state-of-the-art methods, while utilizing a simple paired generator-discriminator architecture, and without leveraging domain-specific knowledge through proxy task design or hand-crafted view consistency objectives driven by data augmentation.

Following reviewer suggestions, we have conducted multiple ablation experiments and comparisons to additional prior work, which we will include in the final version of the paper.  We present these new experimental results here, and respond to individual reviewer questions in separate replies.

### **1. Effect of Data Augmentation**

Contrastive self-supervised learning approaches, including SimCLR (Chen et al., 2020) and InfoMin (Tian et al., 2020), require a carefully calibrated augmentation scheme to achieve decent performance.  For them, both the view consistency objective, which pushes differently augmented variants of the same input to have the same representation, and a well-crafted augmentation scheme are essential.  Though we adopted some minimal data augmentation in our experiments, our approach is far less sensitive to data augmentation than contrastive learning and can learn even without any data augmentation.  We compare our method and SimCLR under varying augmentation schemes to quantitatively showcase these advantages.

We experiment on CIFAR-10/100 and report the results in Tables 1 and 2. In Data Aug, we adopt the following notation for data augmentation: *F* as randomly horizontal flipping, *C* as randomly image cropping, *J* as color jittering, and None as no data augmentation applied during training.  For the same augmentation scheme, our method demonstrates a clear advantage over SimCLR across all evaluation metrics, and moreover, is able to operate even without data augmentation -- a regime in which SimCLR fails.

&emsp; **Table 1: Data Augmentation Ablation on CIFAR-10 Dataset.**

| Data Aug  |  Method      | KMeans    | SVM       | Linear Probing  | 1/5 KNN|
| :---:     |    :----:    | :----:    | :----:    | :----:          | :----: |
| None| Ours|**76.50**| **84.55**|**83.27**| **79.77**/**82.27** |
|None| SimCLR|14.26|21.88|21.85|14.76/15.17|
| ---       | ---          | ---       | ---       | ---             | --- |
| F         |Ours          | **76.29** | **85.74** | **85.83**       | **80.99**/**84.52** |
| F         |SimCLR        | 17.56     | 32.83     | 32.18           | 23.38/25.59 |
| ---       | ---          | ---       | ---       | ---             | --- |
|F + C      |Ours          | **80.08** | **89.34** | **88.41**       | **87.76** /**89.28** |
|F + C      |SimCLR        | 27.87     | 72.76     | 72.29           | 64.93 /67.25 |
| ---       | ---          | ---       | ---       | ---             | --- |
|F + C + J  |SimCLR        | 78.02     | 90.7      | 90.24           | 88.18 /89.57 |


&emsp; **Table 2: Data Augmentation Ablation on CIFAR-100 Dataset.**

| Data Aug  | Method       | KMeans    | SVM       | Linear Probing  | 1/5 KNN|
| :---:     |    :----:    |  :---:    | :----:    | :----:          | :----: |
| None      |Ours          | **22.54** | **52.21** | **51.67**       | **37.54**/**37.46** |
| None      |SimCLR        | 2.18      | 4.59      | 5.49            | 2.72/2.55 |
| ---       | ---          | ---       | ---       | ---             | --- |
| F         |Ours          | **30.80** | **52.05** | **56.50**       | **41.03**/**42.99** |
| F         |SimCLR        | 4.16      | 10.38     | 10.92           | 6.04/5.54 |
| ---       | ---          | ---       | ---       | ---             | --- |
|F + C      | Ours         | **37.41** | **63.21** | **62.08**       | **55.00**/**56.12** |
| F + C     |SimCLR        | 12.20     | 35.14     | 34.36           | 30.83 /29.15 |
| ---       | ---          | ---       | ---       | ---             | --- |
| F + C + J |SimCLR        | 41.79     | 65.29     | 62.26           | 59.28 /61.48 |

---

> ### Author Response · Authors · 2022-11-19
> **Response to all Reviewers (2/2)**
>
> ### **2. Comparison to Other Generative Feature Learning Methods**
>
> (1) **GenRep** (Jahanian et al., 2021) distills the latent space of a pre-trained BigBiGAN (Donahue et al., 2019) model using contrastive objectives, where they assume the adjacent sample in the generator's latent space suffices as a positive sample.  To compare with this method, we train our approach on the ImageNet-100 dataset and mostly follow the training configuration in our ImageNet-10 experiment, except we adopt the BigGAN-deep generator and we extend the training epochs to 1000. For a fair comparison, we follow GenRep's evaluation protocol on linear probing.
>
> &emsp;&emsp; &emsp;&emsp;&emsp;&emsp;&emsp;&emsp;&emsp;&emsp;&emsp;&emsp;&emsp;&emsp;**Table 3: Comparison to GenRep.**
>
> | Method             | Feature Network | View-consistency   | Linear Probing |
> | :---:              |    :----:       |         :---:      | :----: |
> | Ours               | ResNet-18       | No                 | **59.9** |
> | GenRep (Tz only)   | ResNet-50       | No                 | 55.0* |
> *Result is from Figure 6 of GenRep (Jahanian et al., 2021)
>
> Our method outperforms GenRep on ImageNet-100 though we adopt a simpler feature learner (ResNet-18 vs. ResNet-50) and a more direct training pipeline (conventional GAN pipeline vs. extra training loop on generative images).
>
> (2) **Self-condition GAN** (Liu et al., 2020a) clusters the discriminator's features iteratively in a self-discovering fashion, and cluster information is fed to the conditional-GAN pipeline as conditional input.
>
> Though this method produces clusters during training, its objective differs entirely from ours. Their motivation is improving the diversity of image generation rather than representation learning, and they do not include explicit training objectives to enhance feature learning capability.
>
> For comparison, we use their metric to report our model's performance on the CIFAR-10 dataset. The following table shows that our method significantly improves over self-conditioning GAN.
>
> **Table 4: Comparison to self-condition GAN.**
>
> | Method             | NMI          | Purity |
> | :---:              |    :----:    | :---:  |
> | Ours               | **72.77**    | **81.52** |
> | Self-condition GAN | 33.26        | 11.73 |
>
>
> ### **3. Ablating with Other GAN Variants**
>
> (1) **Feature Matching** (Salimans et al., 2016) introduces extra perceptual loss style objectives on the GAN pipeline, where the features for the loss are from the discriminator rather than a pre-trained encoder.  In our implementation, we extract the feature map from the output of all 4 ResNet blocks. The feature matching target is implemented as L1 distance on the averaged featured map between true and generative samples and added to G's training targets.  From the results, there is no clear benefit of using feature matching.
>
> &emsp;&emsp;&emsp;&emsp;&emsp;   **Table 5: Ablation on Feature Matching Loss.**
>
> | Dataset   | Feature Matching   | KMeans          | SVM       | Linear Probing  | 1/5 KNN |
> | :---:     |    :----:          | :----:          | :----:    | :----:          | :----:  |
> | CIFAR-10  | No                 | **80.08**       | 89.34     | 88.41           | **87.76** /**89.28** |
> | CIFAR-10  | Yes                | 77.70           | **89.83** | **88.75**       | 87.30/89.18 |
> | CIFAR-100 | No                 | **37.41**       | **63.21** | **62.08**       | **55.00**/**56.12** |
> | CIFAR-100 | Yes                | 37.07 |     61.67      |  60.88               | 54.45/55.05 |
>
> (2) **FastGAN** (Liu et al., 2020b).  We ablate the learned representation with different generators.  Since FastGAN doesn't provide the generator for 32x32 resolution, we modify FastGAN's generator for 128x128 resolution by removing the upsampling in both feat 64 and feat 128 blocks.  We compare our default BigGAN generator with the same training hyperparameter for CIFAR-10/100 dataset and report the result in Table 6.  FastGAN has slightly worse performance than BigGAN.
>
> &emsp;&emsp;&emsp;&emsp; &emsp; **Table 6: Ablation on FastGAN's generator.**
>
> |Dataset    | Generator | KMeans    | SVM       | Linear Probing  | 1/5 KNN |
> | :---:     |    :----: | :----:    | :----:    | :----:          | :----:  |
> | CIFAR-10  | BigGAN    | **80.08** | **89.34** | **88.41**       | **87.76** /**89.28** |
> | CIFAR-10  | FastGAN   | 80.03     | 87.24     | 86.98           | 86.07/87.66 |
> | CIFAR-100 | BigGAN    | 37.41     | **63.21** | **62.08**       | **55.00**/**56.12** |
> | CIFAR-100 | FastGAN   | **37.46** | 61.14     | 59.25           | 53.95/54.84 |

---

> > ### Author Response · Authors · 2022-11-19
> > **Response to all Reviewers (References)**
> >
> > [1] Chen, T., Kornblith, S., Norouzi, M., Hinton, G. (2020, November). A simple framework for contrastive learning of visual representations. In International Conference on Machine Learning (pp. 1597-1607). PMLR.
> >
> > [2] Tian, Y., Sun, C., Poole, B., Krishnan, D., Schmid, C., Isola, P. (2020). What makes for good views for contrastive learning?. Advances in Neural Information Processing Systems, 33, 6827-6839.
> >
> > [3] Liu, S., Wang, T., Bau, D., Zhu, J. Y., Torralba, A. (2020a). Diverse image generation via self-conditioned GANs. In Proceedings of the IEEE/CVF Conference on Computer Vision and Pattern Recognition (pp. 14286-14295).
> >
> > [4] Liu, B., Zhu, Y., Song, K., Elgammal, A. (2020b, September). Towards faster and stabilized GAN training for high-fidelity few-shot image synthesis. In International Conference on Learning Representations.
> >
> > [5] Jahanian, A., Puig, X., Tian, Y., Isola, P. (2021, September). Generative Models as a Data Source for Multiview Representation Learning. In International Conference on Learning Representations.
> >
> > [6] Salimans, T., Goodfellow, I., Zaremba, W., Cheung, V., Radford, A., Chen, X. (2016). Improved techniques for training GANs. Advances in Neural Information Processing Systems, 29.
> >
> > [7] Donahue, J., Krähenbühl, P., & Darrell, T. (2016). Adversarial Feature Learning.
> >
> > [8] Donahue, J., & Simonyan, K. (2019). Large scale adversarial representation learning. Advances in neural information processing systems, 32.

---

### Decision · Program_Chairs · 2023-01-20

**Decision:**

Reject

**Justification For Why Not Higher Score:**

Contributions are not well positioned within the literature: clustering GAN literature is disregarded under the claim that the paper's goal is to perform representation learning from a GAN discriminator, yet the paper proposes to train the discriminator by leveraging clustering losses, raising the question whether the discriminator of any clustering GAN model could be used out-of-the-box and achieve comparable/better results. The proposed approach is probed from a representation learning perspective and compared against contrastive SSL methods. However, the experimental protocol is not standard, making the claims in the paper unconvincing.

**Justification For Why Not Lower Score:**

N/A

**Metareview: Summary, Strengths And Weaknesses:**

This paper was reviewed by three knowledgeable referees. The reviews highlighted the relevance of the proposed approach but raised important concerns w.r.t. the positioning of the work within the literature (XWaT, 9r54, GyrJ), and the evaluation of the proposed approach - in terms of comparisons, as well as unconventional evaluation protocol and/or dataset choices (XWaT, 9r54, GyrJ). The authors actively engaged in discussion with the reviewers. After discussion, the reviewers remain unconvinced and reiterate their concerns w.r.t. the positioning of the contributions and experiments. The authors ask the AC to take an in-depth look to their discussion with the reviewers as they believe there might have been a mischaracterization of their work. The AC takes into consideration the authors request during the discussion with reviewers. After going over the paper and the reviews, the AC notes the following:
1. The paper emphasizes in several places that the proposed approach is competitive in terms of image generation quality (this appears in the abstract already).
2. The proposed approach is based on the GAN framework and does leverage a clustering loss to improve the representation learning capabilities of the discriminator. Although clustering GAN approaches might have had a different motivation when introduced (e.g. improving image generation as opposed to improving representation learning), they do seem very reasonable baselines to consider as the representations learned by their discriminators might be useful in downstream tasks. The motivation of introducing the proposed new GAN-based approach would be strengthened by showing it is effective when compared to already existing methods which could benefit the same problem. Even if those methods were introduced in a different context, comparing against them (from the perspective of representation learning) is important to justify the contributions of this paper.
3. The evaluation of the proposed approach (again, from a representation learning perspective) is not standard. The authors claim their work should be seen as pioneering and does not need large scale validation, yet using unconventional training and evaluation protocols makes it really hard to assess the value of the contribution. For example, if the baselines have been retrained to follow a different experimental protocol, then it is unclear whether these baselines were given the same opportunity to shine (i.e. was there sufficient hyper-parameter search to make performance claims).
4. If the authors would like to make claims w.r.t. the quality of the generations achieved by their proposed approach, then they should consider baselines beyond BigGAN (Table 2) for that. In this case, a comparison with clustering GAN approaches becomes even more important, as those are very natural baselines to compare the proposed framework against in terms of image generation quality.

Unfortunately, the authors chose to not include those comparison results during the discussion phase.

Therefore, the AC agrees with the reviewers' assessment and recommends to reject. The AC strongly encourages the authors to consider the reviewers' feedback to rethink the presentation of their work (what key messages they would like to emphasize) and improve its execution to make it more convincing.